# ECONGROWTHAGENT: ECONOMIC GROWTH SIMULATION BASED ON LLM AGENTS AND GROWTH THEORY

## ABSTRACT

Economics has long sought to understand economic phenomena to enrich human society, yet it has been constrained by being "a discipline where experiments cannot be conducted" due to ethical and practical limitations. While Agent-Based Models (ABM) offer a computational alternative, modeling complex human decision-making remains fundamentally challenging. Recent large Language Models (LLMs) provide new capabilities for this modeling. We present **Econ-GrowthAgent**, the first LLM-based ABM that simulates economic growth—the most essential phenomenon in economics. Our approach decomposes macroeconomic growth theory into a micro-level dynamic model, where decisions of LLM-based economic agents interact and evolve. Through 25-year simulations with 100 agents, we demonstrate that EconGrowthAgent reproduces economic growth and related key macro phenomena while enabling seamless micro-to-macro analysis. We further demonstrate its value by simulating a counterfactual scenario difficult to explore in reality—an approaching civilization-ending asteroid. As a "laboratory for economic experiments," EconGrowthAgent advances our understanding of economic phenomena and propels economics forward.

## 1 INTRODUCTION

Economics has sought to understand economic phenomena to enrich human society. Microeconomics analyzes decision-making by economic agents such as consumers and firms. Typical examples include consumers' decisions on purchase quantities and firms' decisions on pricing and investment. Macroeconomics directly models relationships between aggregate (*macro*) economic variables, bypassing the modeling of individual economic agents. A representative example is the "Phillips curve" (Phillips, 1958), an inverse correlation between inflation and unemployment.

The ideal way to understand economic phenomena is to reproduce them experimentally and analyze the results in detail. This mirrors how natural sciences investigate phenomena through laboratory experiments. In economics, however, phenomena arise from complex interactions within human society, requiring experiments in society itself. Such experiments—treating society as a "laboratory"—are ethically unacceptable and practically infeasible. Thus, economics is often described as a discipline where experiments cannot be conducted (Samuelson, 1948; Friedman, 1953).

Meanwhile, agent-based models (ABMs) (Gilbert, 2008b), developed in computer science, offer a computational alternative for economic experiments. ABMs conduct computer-simulated experiments using programs (*agents*) that model real-world economic agents. Free from ethical and practical constraints, experiments can be conducted flexibly. Importantly, ABMs enable seamless analysis from micro-level decisions of agents to emerging macro-level phenomena (Epstein & Axtell, 1996).

However, applying ABMs to economics faces a fundamental difficulty. Human decision-making is complex—context-dependent and multifaceted (Groeneveld et al., 2017). For example, consumers typically reduce purchases when prices rise, yet may increase spending despite higher prices after positive events like a favorite sports team's win (Kessler et al., 2022). Enumerating such patterns for rule-based agents is infeasible, leaving economic agent modeling inherently incomplete.

Recently large language models (LLMs) have the potential to address this challenge. Trained on large-scale text, LLMs can approximate patterns of human decision-making (Horton, 2023b; Xie et al., 2024); ABMs that use LLMs as agents—which we term *LLM-ABMs*—should genuinely reproduce real-world economic phenomena.

In this study, we develop an LLM-ABM to simulate *economic growth*. Economic growth refers to the increase in goods production, usually measured by GDP. It drastically improves people's living

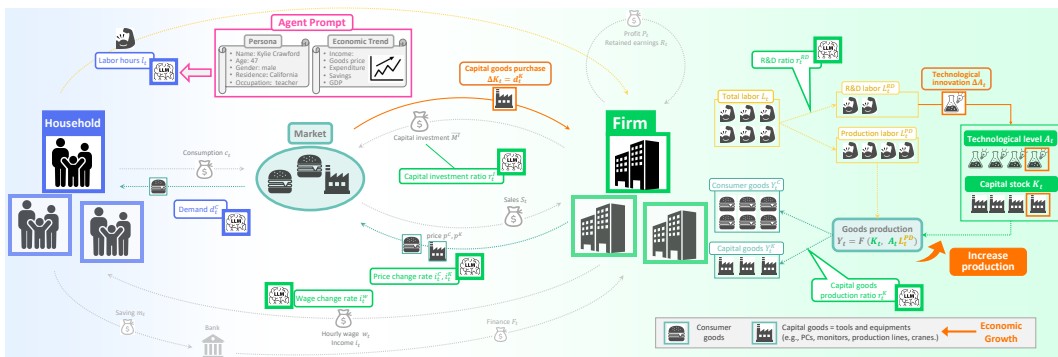

Figure 1: EconGrowthAgent simulates *economic growth* emerging from the interactions between *households* and *firms*, both modeled as LLM agents. LLMs given the agent prompt approximate real-world decision-making. Their decisions and economic variables interact and evolve within our dynamic model.

standards; indeed, global GDP has expanded about 20-fold over the past century (Maddison, 2001), improving food, leisure, healthcare, and education. Thus, understanding the underlying mechanism of economic growth has been particularly emphasized in economics (Smith, 1776; Jr., 1988).

Realizing economic growth with LLM-ABM is highly challenging. First, prior LLM-ABM Li et al. (2024a) modeled only consumers (*households*) and reproduced macro phenomena, specifically, the Phillips curve and Okun's law (Okun, 1962). However, to reproduce economic growth, we must also model *firms*, as they produce goods. Firms' strategic decision-making—spanning investment, R&D, wage setting, pricing, and product-portfolio choices—is far more complex than households', which centers on labor supply and consumption. Next, economic growth arises from complex *dynamic interactions* between households and firms, such as: " ...strong household demand prompts firms to invest and expand capacity. As the labor market tightens and overtime becomes widespread, wage pressures raise unit labor costs and squeeze firms' finances. Firms then scale back investment, and the initial boost to growth slows ...". However, since earlier work considered only narrow fragments of these dynamics tractable for rule-based agents (related work in Section A.1), a comprehensive analysis of economic growth dynamics remains elusive.

We first turn to *economic growth theory* for insights into the drivers of growth (Section 2). Growth theory attributes growth to increases in production equipment and improvements in production technology. However, as a macroeconomic framework, it does not specify the micro-level mechanisms that determine these macro-level factors. We therefore break down growth theory into a novel micro-level dynamic model, which comprehensively specifies how the decisions of economic agents and economic variables interact and evolve (Section 3).

We then implement LLM agents to drive these dynamics with human-like decisions. We provide LLMs with *agent prompts*, specifying their roles (household or firm) and personas (e.g., age, gender, occupation) to reflect real-world diversity. To encourage rational decisions, we also provide economic trends—variables including income, prices, demand, and supply. Integrating our dynamic model and LLM agents finally yields a simulation environment named **EconGrowthAgent** (fig. 1).

We conduct 25-year simulations with 100 agents modeled by frontier LLMs. We first confirm that EconGrowthAgent can reproduce economic growth along with related key macroeconomic phenomena (Section 5). We then analyze underlying micro-level decision-making and verify that it approximates real-world decision-making, particularly firms' strategies such as "raising prices under excess demand" and "shifting production toward high-demand goods." Overall, these analyses demonstrate seamless micro-to-macro analysis and validate EconGrowthAgent from both perspectives.

Next, to emphasize EconGrowthAgent's value as a "laboratory for economic phenomena," we simulate a counterfactual scenario difficult to explore in reality, specifically "a forecast of an approaching civilization-ending asteroid" (Section 6). Results show that households abandon work to spend final moments with loved ones, while firms discard long-term strategies such as investment and R&D, causing economic collapse. Such human-like decision-making is uniquely possible with LLMs.

EconGrowthAgent serves as an extensible foundation for LLM-ABMs. We conclude with an extension guide to catalyze future research, including policy simulations, game-theoretic simulations, and multi-country trade, among others (Section 7). We summarize our contribution as follows:

- We propose EconGrowthAgent[1], the first LLM-ABM capable of simulating economic growth—the most critical phenomena in economics that enrich human society. As a laboratory for economic experiments, EconGrowthAgent advances our understanding of economic growth.
- To achieve EconGrowthAgent, we propose LLM-based modeling of economic agents and a novel dynamic model—a comprehensive micro-level breakdown of economic growth theory—where agents' decisions and economic variables interact and evolve.
- Through long-term simulations using numerous agents, we verify that EconGrowthAgent can reproduce economic growth and related key phenomena, and that the underlying micro-level decision-making approximates that of real-world humans. These micro-to-macro analyses validate EconGrowthAgent from both perspectives.
- We demonstrate that EconGrowthAgent can freely investigate counterfactual scenarios that are difficult to explore in reality.
- EconGrowthAgent serves as an extensible foundation for future LLM-ABMs; we demonstrate this by showcasing concrete examples of future directions.

## 2 PRELIMINARY - ECONOMIC GROWTH THEORY

### 2.1 PRODUCTION FUNCTION

Let $Y$ denote the quantity of goods produced in a given period, which corresponds to GDP and indicates a country's prosperity. $Y$ is determined by the quantity of labor $L$ and the capital stock $K$. $L$ is the sum of workers' labor hours. $K$ refers to the accumulated quantity of *capital goods*—equipment used in production, from office items (desks, PCs) to industrial facilities (production lines, cranes). The relationship between these variables is modeled by a *production function* $\mathcal{F}$:

$$Y = \mathcal{F}(K, L) \tag{1}$$

For simplicity, we consider the Cobb-Douglas function:

$$\mathcal{F}(K, L) = F^0 \times \left(\frac{K}{K^0}\right)^\alpha \left(\frac{L}{L^0}\right)^{1-\alpha}, \ \alpha \in (0, 1) \tag{2}$$

where $F^0$, $K^0$, and $L^0$ are normalization constants. The Cobb-Douglas function satisfies some important properties (Section D.1), such as monotonically increasing in $K$ and $L$.

### 2.2 CONSUMER GOODS AND CAPITAL GOODS

A country produces consumer goods, such as food, clothing, in addition to capital goods. Assuming a fixed proportion $c \in [0, 1]$ consists of consumer goods $C = cY$, the capital goods quantity $I$ is:

$$I = Y - C = (1 - c)Y \stackrel{\text{def}}{=} sY \tag{3}$$

While consumer goods are consumed within the production period, capital goods can be reused in future periods, i.e., they are made for future investment. So, $s \stackrel{\text{def}}{=} 1 - c$ is called *investment ratio*.

### 2.3 CAPITAL STOCK ACCUMULATION DRIVES ECONOMIC GROWTH

We describe the time evolution of the economy by introducing discrete time $t$:

$$\boxed{\textbf{Solow–Swan} \text{ (Solow, 1956b; Solow, 1956)} \quad Y_t = \mathcal{F}(K_t, L_t) \quad I_t = sY_t \quad K_{t+1} = K_t + I_t} \tag{4}$$

The third equation states that the next period's capital stock $K_{t+1}$ increases by investment $I_t$. Assuming constant labor ($L_{t+1} = L_t$) and noting that $\mathcal{F}$ is monotonically increasing in $K$, next period's production strictly increases whnever $s > 0$: $Y_{t+1} = \mathcal{F}(K_{t+1}, L_{t+1}) = \mathcal{F}(K_{t+1}, L_t) > \mathcal{F}(K_t, L_t) = Y_t$. Hence, the Solow–Swan model explains economic growth through *capital stock accumulation via investment*.

### 2.4 TECHNOLOGICAL INNOVATION DRIVES ECONOMIC GROWTH

In reality, production can vary even with the same capital stock due to improved equipment, better processes, and accumulated expertise. We aggregate these factors into a single "technology level" $A_t$, then extend eq. (4) as:

$$\boxed{\textbf{Romer} \text{ (Romer, 1986b)} \quad L_t = L_t^{PD} + L_t^{RD} \quad Y_t = \mathcal{F}(K_t, A_t L_t^{PD}) \quad A_{t+1} = A_t + \gamma L_t^{RD} A_t^\phi} \tag{5}$$

The first equation allocates labor $L_t$ between production sector $L_t^{PD}$ and R&D $L_t^{RD}$. Assuming a *constant R&D share* $r_0$, we have $L_t^{PD} = (1 - r_0)L_t$ and $L_t^{RD} = r_0 L_t$. The second equation states that only the production labor $L_t^{PD}$ is used in production $Y_t$. $Y_t$ depends also on "technology level" $A_t$, which grows through R&D $L_t^{RD}$ as in the third equation (where $\gamma$ and $\phi$ are constants). Hence, the Romer model explains economic growth via *technological innovation through R&D*.

---

[1]Code available at https://anonymous.4open.science/r/econ-growth-agent-D734/README.md

## 3 ECONGROWTHAGENT

Economic growth theory explains growth via two factors—capital stock accumulation through investment and technological innovation through R&D—assuming *given and fixed ratios* for investment ($s$) and R&D ($r_0$). However, in reality, these factors arise from interactions among micro-level economic agents. Without this micro-level lens, a comprehensive analysis of growth is impossible.

We therefore break down growth theory into a micro-level dynamic model (i.e., a system of equations). This model describes how agents' decisions and economic variables interact and evolve. Capital accumulation is modeled via firms that produce and sell capital goods, while other firms purchase them for future production. Technological innovation is modeled via firms allocating a share of labor to R&D, which raises their technology levels. Beyond these investment and R&D decisions, economic growth is also influenced by other decisions made by firms and by households. For instance, funds for purchasing capital goods come from previous sales, which in turn depend on firm-set prices, competitors' prices, and household demand. Thus, we enumerate key decision items for both firms and households together with the relevant interactions.

Then, we implement LLM agents to drive these dynamics with human-like decisions by households and firms, yielding an executable simulation environment named **EconGrowthAgent** (fig. 1).

### 3.1 LLM AGENTS

Assume we have $N^H$ household agents. Each household $h \in \{1, \ldots, N^H\}$ is assigned a random *persona prompt* $\mathcal{P}r^{(h)}$ to express real-world diversity, such as:

> You are Kylie Crawford, a 47-year-old individual living in San Jose, working as a high school teacher ...

It also receives a *decision prompt* $\mathcal{D}^H$ (fig. E.5) listing decision items (e.g., "How many goods to buy?") and an *economic trend prompt* $\mathcal{E}_t^{(h)}$ providing past economic variables (fig. E.7a), up to timestep $t$ (month), to support rational decisions. Examples include income, goods prices, and expenditure. We concatenate these into a single *agent prompt* $\mathcal{P}_t^{H(h)} = \mathcal{P}r^{(h)} \oplus \mathcal{D}^H \oplus \mathcal{E}_t^{(h)}$.

Likewise, each firm $f \in \{1, \ldots, N^F\}$ receives $\mathcal{P}_t^{F(f)} = \mathcal{P}r^{(f)} \oplus \mathcal{D}^F \oplus \mathcal{E}_t^{(f)}$, where $\mathcal{D}^F$ includes the firm's decision items (fig. E.6) such as goods price and investment, and $\mathcal{E}_t^{(f)}$ includes past firm-related variables (fig. E.7b) such as demand and sales. See section E for details of these prompts.

### 3.2 HOUSEHOLD DYNAMICS

▶ **Labor and Income:** The household $h$ determines their working hours. This labor is provided to its employer firm (described later), and the household receives income for the month $t$:

$$\text{labor hours } l_t^{(h)} = \mathcal{LLM}(\mathcal{P}_t^{H(h)}) \qquad \text{monthly income } i_t^{(h)} = w_t^{(h)} \times l_t^{(h)} \tag{6}$$

$w_t^{(h)}$ is the hourly wage. LLM decisions are shown in blue.

▶ **Consumption and Savings:** The household determines the quantity of consumer goods to purchase (*demand*), and purchases from the firm $\sigma_t^H = \arg\min_f p_t^{C(f)}$ offering the lowest price $p_t^{C(f)}$:

$$\text{demand for consumer goods } d_t^{C(h) \to (\sigma_t^H)} = \mathcal{LLM}(\mathcal{P}_t^{H(h)}) \text{ expenditure } c_t^{(h)} = d_t^{C(h) \to (\sigma_t^H)} \times p_t^{C(\sigma_t^H)} \tag{7}$$

After paying for the company, the household deposits the remainder in the bank (Section F.1.1).

### 3.3 FIRM DYNAMICS

Firm decision variables are shown in green.

▶ **Labor Allocation Between Departments:** The firm $f$ allocates labor from its employees $E^{(f)} \subset \{1, \ldots, N^H\}$, a random subset of households, between production and R&D department:

$$\text{total labor } L_t^{(f)} = \sum_{h \in E^{(f)}} l_t^{(h)} \qquad \text{R\&D ratio } r_t^{RD(f)} = \mathcal{LLM}(\mathcal{P}_t^{F(f)}) \tag{8}$$

$$\text{R\&D labor } L_t^{RD(f)} = r_t^{RD(f)} \times L_t^{(f)} \qquad \text{production labor } L_t^{PD(f)} = (1 - r_t^{RD(f)}) \times L_t^{(f)} \tag{9}$$

▶ **Goods Production by Production Department:** The total production capacity of the firm is constrained by the Cobb-Douglas function $\mathcal{F}$ eq. (2). From that capacity, the firm produces consumer and capital goods, whose ratio is decided by the LLM:

$$\text{total production } Y_t^{(f)} = \mathcal{F}(K_t^{(f)}, A_t^{(f)} L_t^{PD(f)}) \quad \text{capital goods production ratio } r_t^{K(f)} = \mathcal{LLM}(\mathcal{P}_t^{F(f)}) \tag{10}$$

$$\text{consumer goods production } Y_t^{C(f)} = (1 - r_t^{K(f)}) \times Y_t^{(f)} \quad \text{capital goods production } Y_t^{K(f)} = r_t^{K(f)} \times Y_t^{(f)}$$

Here, $K_t^{(f)}$ is firm $f$'s capital stock, and $A_t^{(f)}$ its technology level.

▶ **Technological Innovation by R&D Department:**

$$A_{t+1}^{(f)} = A_t^{(f)} + \Delta A_t^{(f)} \tag{11}$$

The growth $\Delta A_t^{(f)}$ depends on R&D labor $L_t^{RD}$ (Section F.2.1).

▶ **Setting Prices of Goods:**

consumer goods price change rate $i_t^{C(f)} = \mathcal{LLM}(\mathcal{P}_t^{F(f)})$  price change $p_{t+1}^{C(f)} = (1 + i_t^{C(f)}) \times p_t^{C(f)}$

capital goods price change rate $i_t^{K(f)} = \mathcal{LLM}(\mathcal{P}_t^{F(f)})$  price change $p_{t+1}^{K(f)} = (1 + i_t^{K(f)}) \times p_t^{K(f)}$

▶ **Sales of Goods:** The firm sells consumer goods to households and capital goods to other firms:

$$\text{consumer goods sales } S_t^{C(f)} = p_t^{C(f)} \times D_t^{C(f)} \quad \text{demand } D_t^{C(f)} = \sum_{h \in [1, N^h]} d_t^{C(h) \to (f)} \tag{12}$$

$$\text{capital goods sales } S_t^{K(f)} = p_t^{K(f)} \times D_t^{K(f)} \quad \text{demand } D_t^{K(f)} = \sum_{f' \in [1, N^F]} d_t^{K(f') \to (f)} \tag{13}$$

The goods inventories are updated depending on the sales volume (Section F.2.2).

▶ **Wage Revision:** The firm can revise employees' wages, e.g., to boost motivation or reduce cost:

wage change rate $i_t^{w(h)} = \mathcal{LLM}(\mathcal{P}_t^{F(f)})$  wage change $w_{t+1}^{(h)} = (1 + i_t^{w(h)}) \times w_t^{(h)}$ $(h \in E^{(f)})$

▶ **Wage Payment:** The firm pays employee wages from sales plus bank financing $F_t$ (Section F.3):

$$\text{total sales } S_t^{(f)} = S_t^{C(f)} + S_t^{K(f)} \quad \text{personnel cost } C_t^{(f)} = \sum_{h \in E^{(f)}} w_t^{(h)} \times l_t^{(h)}$$

$$\text{remaining funds } \overline{M}_t^{(f)} = S_t^{(f)} + F_t^{(f)} - C_t^{(f)}$$

▶ **Capital Stock Accumulation by Capital Goods Purchase:** The firm can purchase capital goods to expand future production (*capital investment*). Within its budget of $\overline{M}_t^{(f)}$ plus retained earnings $R_t^{(f)}$ from the previous periods, the firm buys from the lowest-price firm $\sigma_t^F = \text{argmin}_{f'} \, p_t^{K(f')}$:

capital investment ratio $r_t^{I(f)} = \mathcal{LLM}(\mathcal{P}_t^{F(f)})$  investment funds $\overline{M}_t^{I(f)} = r_t^{I(f)} \times (\overline{M}_t^{(f)} + R_t^{(f)})$ $\quad$ (14)

$$\text{demand for capital goods } d_t^{K(f) \to (\sigma_t^F)} = \lfloor \overline{M}_t^{I(f)} / p_t^{K(\sigma_t^F)} \rfloor \tag{15}$$

The purchased goods are added to the capital stock and used in production from the next period:

$$K_{t+1}^{(f)} = K_t^{(f)} + d_t^{K(f) \to (\sigma_t^F)} \tag{16}$$

Note that the capital goods production ratio $r_t^{K(f)}$ eq. (10) capital investment ratio and $r_t^{I(f)}$ here are completely different. The former determines how many capital goods to produce for sale to other firms, while the latter determines how many to purchase for use in one's own production.

▶ **Profit:** The remaining funds, i.e., profit, become retained earnings for subsequent periods.

$$\text{profit } P_t^{(f)} = \overline{M}_t^{(f)} - \overline{M}_t^{I(f)} \quad \text{retained earnings } R_{t+1}^{(f)} = P_t^{(f)} + R_t^{(f)} \tag{17}$$

## 4 EXPERIMENTAL SETUP

We outline our experimental setup, with additional details provided in Section G.

▶ **Simulation:** We used $N^H$=90 households and $N^F$=10 firms, running 25-year simulations with monthly time steps ($t \in [1, 300]$). Each run required about 110 minutes and cost \$70 in API usage. We ran simulations for five different seeds and show standard deviations.

▶ **Economic Constants:** Parameters were calibrated to match real-world data. Specifically, initial household wages were sampled from the 2024 U.S. wage distribution (U.S. Labor Statistics, 2024), and the capital parameter $\alpha$ in the Cobb–Douglas function eq. (2) was set to 0.3 (Solow, 1956a).

▶ **LLM Agents:** We employed GPT-4.1 (OpenAI, 2023) and Claude-3.5 Sonnet (Anthropic, 2025). The households' economic trend prompt $\mathcal{E}_t^{(h)}$ included 7 economic variables to be referenced during decision-making, while firms' $\mathcal{E}_t^{(f)}$ included 25 variables.

▶ **Decision-Making Analysis.** We analyzed the intermediate reasoning steps generated through chain-of-thought prompting (Wei et al., 2022). See Section E for samples.

## 5 ECONOMIC GROWTH SIMULATION WITH ECONGROWTHAGENT

We first show that EconGrowthAgent can reproduce economic growth (Section 5.1), along with two related key macro phenomena: (i) countries with frugal households grow faster (The World Bank, 1993) (Section 5.2), and (ii) technological innovation enables long-term growth (Romer, 1986a) (Section H.3). For each phenomenon, we analyze the micro-level decision-making of LLMs behind it to test whether it approximates the real-world human behavior. To further ensure that LLMs can approximate firms' strategic decision-making, we conduct statistical analysis of LLMs' decision-making patterns (Section 5.3) and also measure their final business performance (Section H.6).

The results presented below use Claude-3.5 Sonnet as the primary LLM. Section H shows results using GPT-4.1. All phenomena behaved consistently across these two independent LLMs, demonstrating the robustness of our findings. Section H also shows the behavior of all economic variables.

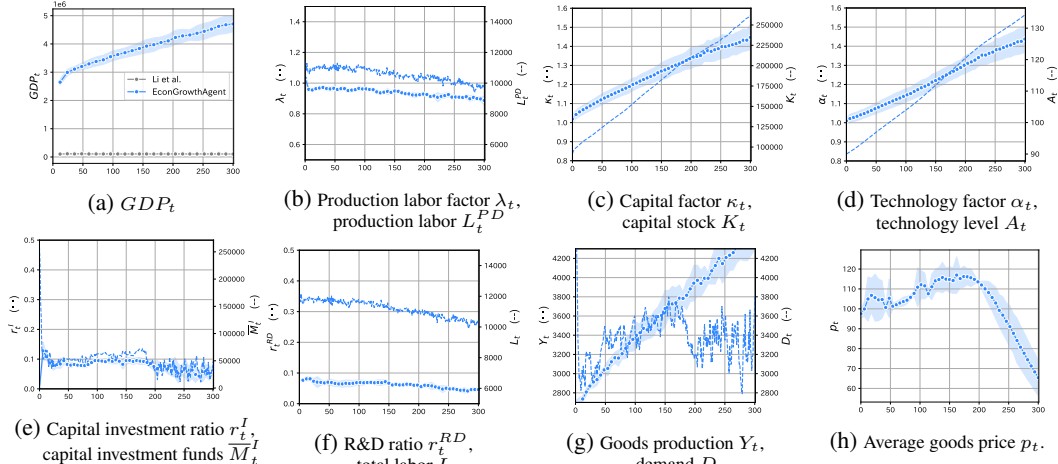

(a) $GDP_t$     (b) Production labor factor $\lambda_t$, production labor $L_t^{PD}$     (c) Capital factor $\kappa_t$, capital stock $K_t$     (d) Technology factor $\alpha_t$, technology level $A_t$

(e) Capital investment ratio $r_t^I$, capital investment funds $\overline{M}_t^I$     (f) R&D ratio $r_t^{RD}$, total labor $L_t$     (g) Goods production $Y_t$, demand $D_t$     (h) Average goods price $p_t$.

Figure 2: 25-year simulation results (timesteps $t \in [1, 300]$ shown on the horizontal axis) using Claude-3.5-Sonnet agents. Each economic variable represents either a *sum or average* across firms (see Section 5.1).

## 5.1 ECONOMIC GROWTH

▶ **Reproduction of the Macro-Level Phenomenon:**
Figure 2a shows GDP over time. $GDP_t = Y_t \times p_{t=1}$ converts the country's total goods production $Y_t$ into monetary value using the base-period price $p_{t=1}$ eq. (H.1). Here, $Y_t$ is the sum of firms' production $Y_t^{(f)}$: $Y_t = \sum_{f \in [1, N^F]} Y_t^{(f)}$. Similarly below, variables without subscripts are either the sum or average of subscripted micro-level variables.

EconGrowthAgent achieved economic growth, with an average annual growth rate of about 2.3%, growing 2-fold over 25 years. In contrast, simulation of Li et al. (2024a) showed nearly constant GDP. A firm's production $Y_t^{(f)}$ eq. (10) can be decomposed into three factors:

$$Y_t^{(f)} = F^0 \times \underbrace{\left(\frac{L_t^{PD}}{L^0}\right)^{1-\alpha}}_{\text{production labor factor } \lambda_t^{(f)}} \times \underbrace{\left(\frac{K_t}{K^0}\right)^{\alpha}}_{\text{capital factor } \kappa_t^{(f)}} \times \underbrace{\left(\frac{A_t}{A^0}\right)^{1-\alpha}}_{\text{technology factor } \alpha_t^{(f)}} \tag{18}$$

Applying the same decomposition to GDP yields $\lambda_t, \kappa_t, \alpha_t$ eq. (H.2). In EconGrowthAgent, $\lambda_t$ remained stable, whereas $\kappa_t$ and $\alpha_t$ increased (figs. 2b to 2d left). These two factors are therefore the main drivers of growth, consistent with growth theory. By design, simulation of Li et al. (2024a) does not incorporate capital stock or technology level, inevitably failing to realize growth. Hereafter, we exclude analysis of Li et al. (2023), which does not offer insights into growth.

▶ **Micro-Level Analysis**
**Dynamics behind a Stable Labor Factor $\lambda_t$:** Households supplied stable labor (fig. 2b, right), consistent with the labor-supply decisions, e.g.,

> *Given that my income has been stable at $7,624 per month, I don't feel a strong need to work significantly more. I will maintain a willingness to work at a similar level as last month . . .*

(Full chain-of-thought excerpts are in fig. E.8.) Such stability aligns with real-world household behavior (Blundell & Macurdy, 1999).

**Dynamics behind a Growing Capital Factor:** Firms set a capital-investment ratio of about 10% (fig. 2e, left), as reflected in decisions such as:

> *The company's profit was $50,649 last month, which is a significant amount. Given the positive sales trend and the need to maintain and expand production capacity, I would recommend investing a substantial portion of the profit, perhaps in the range of 10-20%.*

This range is consistent with typical industry practices (Feldstein, 1987; Doe & Smith, 2022). Resulting purchases of capital goods led to capital-stock accumulation (fig. 2c, right).

**Dynamics behind a Growing Technology Factor:** Firms maintained an R&D ratio around 5-10% (fig. 2f), e.g.,

> *The current allocation of 10% of the labor force to the R&D sector seems reasonable, as it allows for future productivity improvements while maintaining a focus on current production.*

This ratio is in line with commonly observed values (OECD, 2022; Doe & Brown, 2021). Continuous R&D increased technology levels $A_t$ (fig. 2d, right).

**Dynamics behind Supply, Demand and Price:** Production $Y_t$ lagged demand $D_t = D_t^C + D_t^K$ in the early phase ($t \lesssim 150$ in fig. 2g). Firms' pricing decisions responded to excess demand:

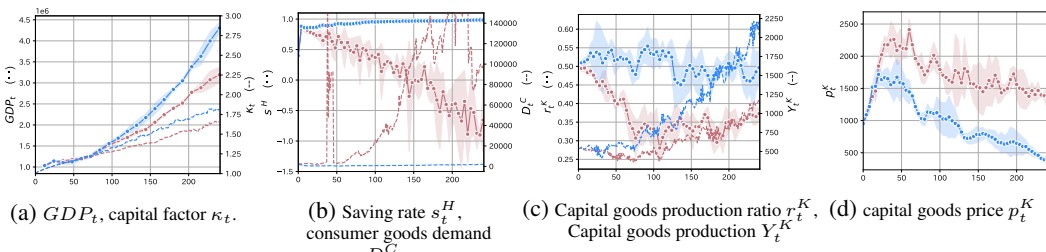

(a) $GDP_t$, capital factor $\kappa_t$.  (b) Saving rate $s_t^H$, consumer goods demand $D_t^C$  (c) Capital goods production ratio $r_t^K$, Capital goods production $Y_t^K$  (d) capital goods price $p_t^K$

Figure 3: A frugal country and a spendthrift country (Section 5.2). We used Claude-3.5-Haiku as the LLM.

> *The demand for consumer goods (91) is significantly higher than the quantity of consumer goods production (20) last month. This suggests that there is room to increase the price of consumer goods without significantly impacting demand. Given the current profit margin, I would recommend a moderate increase in the price of consumer goods, around a 1-3% increase.*

This led to rising prices $p_t$ (fig. 2h left). Such a strategic price adjustment based on market imbalances is consistent with empirically observed market practices (Rotemberg, 1992).

In turn, households' purchase decisions adjusted:

> *Considering my stable income, past consumption of 7 goods, and the increase in luxury goods prices from 126.84 to 150.04, I will slightly reduce my demand for luxury goods this month.*

Accordingly, demand $D_t$ declined as prices rose (fig. 2g, right), in line with empirical evidence (Ito, 2014; Deaton & Muellbauer, 1980). Such demand adjustment aligns with real-world household behavior (Ito, 2014; Deaton & Muellbauer, 1980).

However, later production exceeded demand (fig. 2g) due to the growth, driving down prices (fig. 2h). *More goods at lower prices* signifies improved living standards from economic growth.

**Summary:** EconGrowthAgent reproduces economic growth and enables detailed analysis of underlying micro-level decision-making through LLMs' chain-of-thought traces. Furthermore, this decision-making aligns with real-world human behavior as reported in the existing literature.

### 5.2 Related Key Phenomenon: High Growth in Frugal Countries

▶ **Reproduction of the Macro-level Phenomenon:**
Figure 3 compares growth in a "frugal" vs. a "spendthrift" country. Frugal household agents receive the persona prompt "You feel ultimate joy when your savings increase," while spendthrifts get "You immediately spend your earned money as a reward for yourself." The frugal country's GDP grew faster (fig. 3a left), mainly due to a higher capital factor $\kappa_t$ (fig. 3a right).

▶ **Micro-Level Analysis**
Figure 3b (left) shows the saving rate $s_t^{H}$ [2] , indicating how much of their income households allocate to savings. Frugal households maintained a saving rate near 1.0, whereas spendthrifts sometimes showed negative rates — consuming more than their income and drawing on bank savings. Spendthrift households' purchase decisions was such as:

> *I would like to purchase 150 luxury goods this month, which at the current price of $485 per good, would amount to a total consumption of $72,750 or about 3.36 times my monthly income. This level of consumption is quite high and may not be financially sustainable in the long run. However, I have a strong desire to spend . . .*

Consequently, consumer goods demand $D_t^C$ in the spendthrift country far exceeded that in the frugal country (fig. 3b right).

Figure 3c (left) shows that the capital goods production ratio $r_t^K$ was significantly lower in the spendthrift economy than in the frugal economy, reflecting firms' decisions such as:

> *The demand for consumer goods (2160) is much higher than the demand for capital goods (670). To better meet the consumer demand, I would recommend increasing the proportion of production allocated to consumer goods.*

That is, resources were reallocated from capital to consumer goods to serve their stronger demand, which made capital goods production lower in the spendthrift economy (fig. 3c, right).

As lower supply raises on prices (Section 5.1), the capital goods price $p_t^K$ rose above that in the frugal economy (fig. 3d). So, the firms in the spendthrift country could purchase fewer capital goods (fig. H.11d), hindering capital stock accumulation. This suppressed the economic growth.

---

[2] $s_t^H = \text{avg}_h \left[ (i_t^{(h)} - c_t^{(h)})/i_t^{(h)} \right]$, where $i_t^{(h)}$ is income eq. (6) and $c_t^{(h)}$ is consumption eq. (6).

Table 1: Regression coefficients for firm decisions against referenced economic variables. All variables are *standardized* (zero mean, unit variance). **Red** marks coefficients with both large magnitude and statistical significance. Non-significant coefficients are omitted. See Section H.5 for the detailed setting and full results. $g_t = (D_t - Y_t) / \max(D_t, Y_t)$="demand-supply gap", where $D_t$ is demand and $Y_t$ is production.

| | $\hat{S_{t-1}}$ | $D_{t-1}^C$ | $g_{t-1}^C$ | $D_{t-1}^K$ | $g_{t-1}^K$ |
|---|---|---|---|---|---|
| consumer goods price change rate $i_t^C$ | $0.21_{\pm0.025}$ | $-0.099_{\pm0.013}$ | $0.73_{\pm0.012}$ | $-0.065_{\pm0.014}$ | $-0.035_{\pm0.01}$ |
| capital goods price change rate $i_t^K$ | $0.028_{\pm0.024}$ | $0.09_{\pm0.013}$ | $0.0055_{\pm0.012}$ | $-0.05_{\pm0.014}$ | $0.69_{\pm0.01}$ |
| wage change rate $i_t^w$ | $0.58_{\pm0.038}$ | $0.044_{\pm0.02}$ | $0.24_{\pm0.019}$ | $-0.038_{\pm0.022}$ | $0.19_{\pm0.016}$ |
| capital goods production ratio $r_t^K$ | $-0.19_{\pm0.035}$ | $-0.33_{\pm0.018}$ | $0.052_{\pm0.017}$ | $0.49_{\pm0.02}$ | $0.048_{\pm0.014}$ |
| R&D ratio $r_t^{RD}$ | $0.21_{\pm0.026}$ | $0.1_{\pm0.014}$ | $-0.13_{\pm0.013}$ | $0.16_{\pm0.015}$ | $-0.05_{\pm0.011}$ |
| capital investment ratio $r_t^I$ | $0.01_{\pm0.038}$ | $0.024_{\pm0.02}$ | $0.16_{\pm0.018}$ | $0.1_{\pm0.021}$ | $0.19_{\pm0.016}$ |

Classical economic growth theory (Section 2.3) assumes a given, fixed national investment rate $s$ in capital stocks, and thus cannot explain *why* some countries invest more than others. In contrast, EconGrowthAgent breaks down investment by showing how household frugality versus high spending—via induced changes in firms' production and pricing strategies—generates nation-level investment behavior. That is, it reveals macro phenomena as outcomes of interactions of micro-level decision-making.

## 5.3 STATISTICAL VALIDATION OF FIRM DECISION-MAKING

While Li et al. (2024a) demonstrated that LLMs can model household decision-making, their ability to handle firms' complex strategic decision-making remained unexplored. Analyses in the previous sections provided qualitative evidence that LLMs can model firm behavior. Here, we assess this with greater statistical rigor via regression analyses that characterize their decision-making patterns.

Table 1 shows regression coefficients $\beta_i$ for LLM decisions on referenced economic variables:
$$\text{consumer goods price change rate} i_t^C \approx \beta_1 S_{t-1} + \beta_2 D_{t-1} + \dots$$
We used $30,000$ data points, i.e., $10$ firms $\times 300$ periods $\times 10$ simulation runs (25 years). **Red** marks coefficients with both large magnitude and statistical significance, meaning that LLMs emphasized them in their decision-making.

Our regression analysis reveals three key patterns in firm decision-making:

- **Price adjustments reflecting market imbalances:** consumer goods price change rate $i_t^C$ increased with demand-supply gap $g_{t-1}^C$ [3]. Capital goods price change rate $i_t^K$ followed similar dynamics.
- **Performance-linked Wages:** wage change rate $i_t^w$ correlated with sales trends $\hat{S_{t-1}}$,
- **Responsive production shifts toward high-demand goods:** Capital goods production ratio $r_t^K$ decreased with consumer goods demand $D_{t-1}^C$ and increased with capital goods demand $D_{t-1}^K$.

In contrast, R&D ratio and capital investment ratio $\overline{M}_t^I$ showed no significant correlations with economic variables, as firms maintained these parameters at stable levels, as discussed in 5.1.

These strategic decision-making align with empirically found real-world firm behaviors (Braun et al., 2024; Goolsbee, 1998; Dobbelaere & Mairesse, 2018; Card et al., 2018; Zha, 2024). Furthermore, analysis of business performance confirms that LLM-guided firms achieve sustainable profitability (Section H.6).

## 6 COUNTERFACTUAL SIMULATION WITH ECONGROWTHAGENT

To demonstrate EconGrowthAgent's ability to examine scenarios that are difficult to study in the real world, we simulate a "forecast of an approaching civilization-ending asteroid." We also confirm that LLMs can model the human-like—context-dependent and multifaceted— decision-making.

In the country shown in fig. 4, at $t = 220$ households and firms received the following prompt:

> *Observatories worldwide announced today, "A giant asteroid is approaching Earth with an extremely high probability of collision." According to experts, the asteroid is large enough to cause civilization collapse with immeasurable consequences.*

All other parameters and settings were held constant.

As fig. 4a indicates, GDP fell sharply. This primarily reflects a sharp drop in household labor supply $L_t$ and thus in the production-labor factor $\lambda_t$ (fig. 4b). Household labor decisions included:

> *Given the ongoing significant event, the importance of work may still be diminished as the focus remains on spending time on meaningful activities and with loved ones. I will maintain a reduced willingness to work to allow more leisure time, as the potential end of civilization continues to change the value of accumulating wealth.*

---

[3] $g_{t-1}^C = (D_{t-1}^C - Y_{t-1}^C) / \max(D_{t-1}^C, Y_{t-1}^C)$.

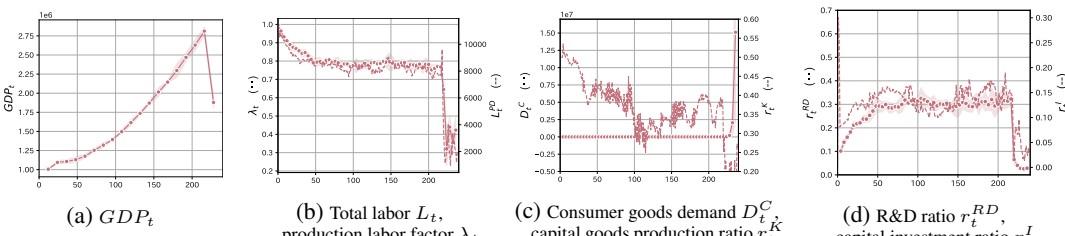

(a) $GDP_t$  (b) Total labor $L_t$, production labor factor $\lambda_t$  (c) Consumer goods demand $D_t^C$, capital goods production ratio $r_t^K$  (d) R&D ratio $r_t^{RD}$, capital investment ratio $r_t^I$

Figure 4: Counterfactual simulation of "forecast of an approaching civilization-ending asteroid" (Section 6).

Additionally, households' demand for consumer goods increased substantially (fig. 4c left):

> *Given the ongoing significant event, I will continue to purchase two luxury goods to enhance my current happiness and make the most of the present moment. I will purchase more luxury goods to continue increasing my current happiness in light of the uncertain future.*

In response to stronger consumer demand, firms reduced their capital goods production ratio (fig. 4c, right), reallocating production resources from capital to consumer goods. Reduced capital goods hindered capital stock accumulation and thereby suppressed the economy, as in Section 5.2.

Finally, both the R&D ratio and the capital-investment ratio dropped sharply (fig. 4d), following:

> *With the impending meteor strike, I do not believe it is worthwhile to allocate labor to the R&D sector, as any long-term innovation would be meaningless. . . . Similarly, I do not believe it is prudent to invest in new capital equipment. The future production capacity is irrelevant given the lack of a long-term future.*

This weakened capital accumulation and technological innovation, further depressing the economy.

As shown above, the *forecast alone* triggered collapse via a chain of agents' decisions.

## 7 EXTENDING ECONGROWTHAGENT: A GUIDE

EconGrowthAgent serves as an extensible foundation for future LLM-ABMs, equipped with fundamental agents and interaction dynamics. Below we present concrete extensions:

- **Policy Simulation:** We can simulate effects of specific policies as external shocks, using the same approach as the "asteroid shock" in Section 6.
- **Game-Theoretic Simulation:** Monopolistic or oligopolistic markets in which a few firms engage in competition. Firms' sales depend on competitors' actions such as pricing and production volume, therefore they must make decisions strategically. Whether theoretically predicted strategies (i.e., Bertrand and Cournot) arise is particularly interesting.
- **Multi-Country Trade** where firms can sell goods to households of other countries. Whether firms in a specific country specialize in specific types of goods, as predicted by international trade theories (Ricardian and Heckscher–Ohlin models), is of interest.
- **Other Agent Types:** Governments defining fiscal policies, tax rates, trade policy. Central banks setting monetary policy.
- **Other Market Types:** Financial markets in which households and firms trade stocks and bonds, and labor markets in which households and firms form employment contracts.
- **The Effect of Agent Diversity on Macro Phenomena:** Households can vary by nationality, religion, and life stage; Firms by strategies, e.g., specialization versus diversification (or conglomerate); Governments by big to small government; Central banks by hawkish versus dovish stances.
- **Agent-to-Agent Communication:** We can examine whether theory-consistent behaviors emerge; Inter-firm communication may lead to collusion, cartels, and mergers and acquisitions; Household-firm communication to work-hour adjustments and wage negotiations.
- **Utility-Maximizing Agents:** We can examine whether explicitly prompting agents to maximize their utility (e.g., leisure for households, profit for firms) optimizes their behaviors.
- **Larger-Scale Economies:** More realistic economies where tens of thousands of agents reside.

## 8 LIMITATIONS

See Section C.

## 9 CONCLUSION

We proposed EconGrowthAgent, an LLM-based ABM for analyzing economic growth—the most critical phenomenon in economics—and empirically validated its efficacy. As a "laboratory for economic experiments," EconGrowthAgent advances our understanding of economic phenomena and propels economics.

## ETHICS STATEMENT

We used Large Language Models (LLMs) to improve grammar and clarity of the paper.

Our study aims to simulate the economy, so it is possible that the simulation results may offer potentially harmful insights. However, we believe that deep understanding of the economy obtained through our simulation will ultimately lead to better society.

## REPRODUCIBILITY STATEMENT

Our code and a step-by-step guide are at `https://anonymous.4open.science/r/econ-growth-agent-D734/README.md` Full details on experimental settings are described in Sections 4 and G.

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

Table A.2: A comparison of our study and previous studies. "–" indicates that the feature is absent. "△" shows simple rule-based decisions, such as "fixed growth rate" ( "Growth Factor – Tech"), "fixed R&D labor ratio," "fixed investment-to-profit ratio," "all household income is consumed," "labor is supplied as much as firms want," "price as cost plus fixed markup," and "fixed wage revision rate."
✓ shows the household-LLMs' decisions whereas ✓ shows firm-LLMs'.

| | Growth Factor | | Firm | | | R&D | Capital | Household | | Market | | | |
|---|---|---|---|---|---|---|---|---|---|---|---|---|---|
| | Capital | Tech | Production | | | | Investment | Consumption | Labor | Cons. goods | Cap. goods | Financial | Labor |
| | | | Cons. goods | Cap. goods | Dual–sector | | | | | Pricing | Pricing | | Wage revision |
| Stolzenburg (2015) (Stolzenburg) | ✓ | △ | ✓ | – | – | – | – | △ | △ | △ | – | – | △ |
| Other Solow (2013) (Lengnick, 2013) | ✓ | △ | ✓ | – | – | – | – | △ | △ | △ | – | – | △ |
| Fagiolo & Dosi (2003) (Fagiolo & Dosi, 2003) | ✓ | ✓ | ✓ | – | – | ✓ | – | – | – | △ | – | – | – |
| Dosi (2010) (Dosi et al., 2010) | ✓ | ✓ | ✓ | ✓ | – | △ | △ | – | – | △ | △ | – | △ |
| **EconGrowthAgent** | ✓ | ✓ | ✓ | ✓ | ✓ | ✓ | ✓ | ✓ | ✓ | ✓ | ✓ | ✓ | ✓ |

# A    RELATED WORK

## A.1    TECHNICAL CONTRIBUTION OF OUR STUDY UPON PREVIOUS STUDIES

Table A.2 compares previous ABMs that addressed economic growth with our work, **EconGrowthAgent**. We only include *non-LLM*-ABMs in the comparison, because no existing studies have explored economic growth using LLM-ABMs.

**Technical Contribution 1: Comprehensive Dynamics for Broader Analysis of Economic Growth:**

The first contribution of EconGrowthAgent is its more comprehensive micro-level dynamics compared to previous ABMs, enabling a broader analysis of economic growth, as demonstrated in our experiments sections 5 to 5.3. More specifically, we highlight the following points.

Economic growth emerges from *complex dynamic interactions* between household and firm decisions. For instance, one can imagine interactions such as "robust consumer demand stimulates firms' investment strategies, accelerating growth." To capture a comprehensive picture of such interaction dynamics, we broke down economic growth theory and enumerated a wide range of decision items for both firms and households, along with all relevant interactions (Section 3).

As a result, EconGrowthAgent can analyze economic growth from more angles than previous ABMs. For example:

- Firms in EconGrowthAgent have dual production sectors for consumer goods and capital goods, with the allocation decided by LLM agents. Hence, we can analyze whether corporate strategies result in the *emergence* of firms that specialize in consumer goods (i.e., B2C) or capital goods (i.e., B2B).

- EconGrowthAgent incorporates R&D and capital investment, allowing us to analyze how these choices affect growth.

- EconGrowthAgent fully models household decision-making, so we can investigate how households' choices (e.g., whether to work more hours) and household–firm interactions (e.g., wage or price negotiations) influence economic growth.

**Technical Contribution 2: Human-like Decision-Making via LLMs for Realistic Simulation:**

Our second contribution is that whereas previous studies used simple rule-based agents (e.g., a fixed R&D ratio or a fixed investment-to-profit ratio; see the caption of Table A.2), we employ LLM-based agents to achieve more realistic, human-like decision-making, as confirmed by our experiments. This approach enables more precise simulation of real-world phenomena. More specifically, we highlight the following points.

Since ABMs derive macro phenomena from micro-level agent interactions, more realistic (i.e., human-like) decisions at the micro level produce more realistic macro-level outcomes. Since LLMs have learned human decision patterns from large-scale text (Horton, 2023b; Xie et al., 2024; Argyle et al., 2023; Fu, 2023; Chen et al., 2023a), we expect them to approximate human agents.

Hence, we incorporated LLMs into our EconGrowthAgent. After confirming that EconGrowthAgent can reproduce economic growth and related key phenomena (Section 5), we verified that LLMs indeed provide human-like decision-making, by examining their chain-of-thought text (sections 5 to 6) and analyzing in detail their decision-making as firm agents (Section 5.3).

In more detail, "human-like" decision-making includes both rational and irrational aspects, and we showed that LLMs can capture *both*. One example of rational behavior is firms' strategic decision-making, such as:

- Firms must determine labor allocation ratios between production and R&D departments. Production addresses immediate demand, while R&D addresses longer-term demand through innovation. Optimal ratios must balance these demands.
- Firms must purchase capital goods to expand future production while maintaining funds for wages and contingencies.
- Firms must determine production ratios that maximize profits between consumer and capital goods.
- Firms must set optimal prices. Higher prices may boost per-unit profit, but risk losing sales in price competition and leaving unsold inventory as customers switch to cheaper competitors.
- Firms must strategically set wages. Higher wages may increase employee motivation and productivity but also raise labor costs.

Obviously, rule-based decision-making cannot address such complex strategic decisions unless specifically hard-coded by designers. Furthermore, since previous LLM-ABMs, such as Li et al. (2024a), considered only simple household decisions (consumption and labor hours), it remained an open question whether LLMs could model these complex and strategic decisions. We answered this question affirmatively by showing that LLMs succeed in firm management (i.e., generating profit) and have rational decision-making patterns such as "raising prices when demand exceeds supply" and "shifting production to goods with higher demand" (Section 5.3).

On the irrational side, we highlight our counterfactual scenario (Section 6) featuring a "forecast of an approaching civilization-ending asteroid," which led households to abandon work to spend time with loved ones and overconsume, acting purely on emotion. Consequently, the *mere prediction* of an asteroid collapsed the economy. Modeling such human-like behavior is uniquely possible with LLM-based agents.

## A.2 ECONOMIC SIMULATIONS WITH CLASSICAL ABMS

Classical ABMs have investigated various phenomena other than economic growth. Pioneering works set the stage by illustrating the micro-to-macro link: individual actors—such as households and firms—behave under bounded rationality and local rules, yet their collective decisions generate complex macroeconomic or social outcomes (Epstein & Axtell, 1996; Tesfatsion, 2006; Gilbert, 2008a). These early studies revealed that ABMs can replicate core empirical regularities such as business cycles, wealth distributions, or segregation patterns without requiring the assumptions of full rationality or representative agents.

Over the past decade, researchers have extended classical ABMs to incorporate richer behavioral models and more realistic institutional contexts (Smith & Lopez, 2020; Dosi et al., 2022). For example, firms may engage in adaptive learning and dynamic pricing strategies, while households face heterogeneous constraints in labor markets or credit access. These refinements enable modeling of phenomena such as technology diffusion, policy interventions, and systemic financial risks in ways that align more with real-world data. Methodologically, modern ABMs benefit from faster computing resources and advanced calibration/validation techniques, making them valuable complements to mainstream quantitative approaches.

## A.3 ECONOMIC SIMULATIONS WITH LLM-ABMS

Following Li et al. (2023), a number of studies have expl/ored Large Language Model (LLM)-based agent simulations from diverse angles. Some have provided broad surveys of this emerging field, reviewing how LLM agents can be integrated into agent-based modeling (ABM) and discussing challenges such as environment perception, alignment, and evaluation (Gao et al., 2024).

Other work has focused on enhancing the realism of LLM-driven economic agents by incorporating preference-driven decision-making and self-learning (Cao et al., 2024). Researchers have also extended simulations to competitive domains, demonstrating that LLM-powered agents' behaviors align with economic theory (Zhao et al., 2024). Meanwhile, large-scale frameworks simulate entire societies with thousands of LLM-based agents to investigate social phenomena, such as polarization and the spread of inflammatory messages (Piao et al., 2025). (Park et al., 2023) demonstrates generative agents that spontaneously coordinate events in a sandbox environment. In finance, LLM-driven trading simulators test price formation and liquidity in realistic market microstructures (Lopez-Lira et al., 2025).

Beyond macro-level analyses, several lines of research have tested the finer-grained fidelity of LLM-based agents' behaviors. For instance, one study examined whether generative agents replicate subtle linguistic norms in hierarchical conversations (Anonymous, 2024), while another used an LLM-based framework to model influence diffusion on social networks through more realistic, personalized messaging (Zhang et al., 2025). In parallel, work on reliability and consistency revealed that LLM-driven simulations sometimes exhibit significant gaps between stated intentions and emergent behaviors (Huang et al., 2024). Novel approaches also employ multiple distinct LLMs to represent heterogeneous socioeconomic groups, capturing more realistic distributions of cognitive traits across agent populations (Xie et al., 2025). Collectively, these studies underline both the promise of LLM-based ABM and the need for more robust methods to ensure behavior authenticity.

Researchers have also investigated how LLM agents handle complex team formation and coordination (Li et al., 2025), as well as systematic evaluations of LLMs' microeconomic reasoning abilities via dedicated benchmarks (Raman et al., 2025). Work on "agent ecosystems" has demonstrated the potential for LLM-based agents to evolve their domain expertise purely from extensive simulation, hinting at a new paradigm for training domain-specific AI (Li et al., 2024b).

Despite growing promise, questions remain regarding validity and reproducibility. Methods to calibrate agent prompts and systematically evaluate LLM biases are active research directions, as are techniques for embedding domain knowledge within LLM-based ABMs (Chen et al., 2023b). Ongoing efforts also explore memory architectures to preserve agent histories, facilitating more stable long-run behaviors (Yamane et al., 2024).

This research stream suggests that harnessing LLMs for ABMs could illuminate complex economic or social processes, while also serving as a complementary platform for rapid experimentation.

### A.3.1 Economic Decision-Making in LLMs

Researchers have begun exploring the use of large language models (LLMs) such as GPT-4 to simulate decision-making processes in economic contexts. An early milestone demonstrated that LLM-based agents, when given suitable prompts, can reproduce human-like outcomes in well-known behavioral experiments, such as the ultimatum game and public goods games (Horton, 2023a; Argyle et al., 2023). This line of research suggests that LLMs have latent "folk economic" reasoning, presumably learned from large text corpora.

Another active thread focuses on forecasting and policy analysis. Some studies investigate whether advanced LLMs can predict inflation, GDP, or central bank actions in ways comparable to established econometric techniques (Fu, 2023; Chen et al., 2023a). While results on pure time-series tasks are mixed, a promising direction leverages LLMs for text-based data interpretation, such as parsing central bank announcements or industry reports, thus complementing rather than displacing standard models. Additional efforts assess how well LLMs conform to game-theoretic rationality or behavioral biases. Evidence indicates that LLMs often replicate core human economic biases, though they can also exhibit inconsistencies depending on the prompt or model variant (Kitadai et al., 2024).

## B    BROADER IMPACTS

Our work leverages large language models (LLMs) to simulate real-world economic decision-making, which can offer tangible benefits to both researchers and practitioners. By approximating households' and firms' choices within a dynamic model, our approach allows users to explore various "what-if" scenarios that are typically difficult or costly to test in reality. This capability may lead to better-informed policy decisions and strategic business insights, potentially reducing the societal and economic risks associated with trial-and-error experiments. Moreover, the ability to examine the micro-level decision-making process through the chain-of-thought text can offer deeper transparency into how and why specific macroeconomic phenomena emerge.

On the other hand, the reliance on LLMs brings potential risks. Model biases embedded in the LLM's training data may skew simulation outcomes, potentially leading to unfair or inaccurate conclusions—particularly with respect to certain populations or regions. Additionally, while the simulation environment captures complex interactions, its results rely on assumptions and approximations that may not fully reflect real-world complexity. Overreliance on such models could inadvertently guide harmful or misguided policy and business decisions. To mitigate these issues, we recommend thorough validation of simulation outputs, continual refinement of LLM-based models, and the use of complementary data sources and methodologies. By doing so, we aim to ensure that our simulation framework serves as a beneficial supplement—rather than a sole authority—for informing economic policy and strategy.

## C    LIMITATIONS

The current version of EconGrowthAgent assumes the following simplifications:

- We chose $N^H + N^F = 100$ agents to balance economic realism with computational costs. See Section G for the details.
- EconGrowthAgent aggregates the entire goods into just two homogeneous goods — consumer goods and capital goods—thereby omitting brand, quality, and other sources of product differentiation. Then, agents always purchase from the lowest-price firm, as price becomes the only factor differentiating goods produced by other firms.
- The model comprises a single closed economy; multinational trade, cross-border capital flows, and exchange-rate dynamics are not yet represented.
- Demographics are static, i.e., the number of agents does not change over time.
- EconGrowthAgent currently omits a government sector, which in reality affects economic growth.

# D  ECONOMIC GROWTH THEORY

## D.1  PRODUCTION FUNCTION

Below are other important properties satisfied by the Cobb-Douglas function.

$$\frac{\partial \mathcal{F}}{\partial K} > 0, \qquad \frac{\partial \mathcal{F}}{\partial L} > 0 \qquad \text{(monotonically increasing)} \tag{D.1}$$

That is, the production increases if $K$ or $L$ increases.

$$\mathcal{F}(zK, zL) = z\mathcal{F}(K, L) \quad \text{(constant returns to scale)} \tag{D.2}$$

That is, when $K$ and $L$ are *simultaneously* increased by a factor of $z$, $Y$ also increases by a factor of $z$. For example, if food processing lines and the number of workers are both doubled, food production doubles as well. Finally:

$$\frac{\partial^2 \mathcal{F}}{\partial^2 K} < 0, \; \frac{\partial^2 \mathcal{F}}{\partial^2 L} < 0 \quad \text{(diminishing returns)} \tag{D.3}$$

Considering the latter, "when increasing labor $L$ while keeping capital stock $K$ *fixed*, the larger $L$ becomes, the smaller the incremental increase in production $\partial \mathcal{F}/\partial L$ from additional $L$" (similarly for the former). If workers are increased without increasing processing lines, efficiency decreases as the number of workers per line increases, so the incremental increase in production from additional workers diminishes.

## D.2  SAVINGS AND INVESTMENT

The uses of produced goods can be classified into three categories: consumption ($C$) by households, investment ($I$) by firms, and government expenditure ($G$) for public purposes:

$$Y = C + I + G \tag{D.4}$$

Goods for household consumption $C$ are called "consumer goods," referring to goods purchased by general consumers such as the authors, including food, clothing, and entertainment. Goods for firm investment $I$ are capital goods, purchased to increase future production. Government expenditure $G$ includes costs for compulsory education and social security, which are also classified as consumer goods.

Equation (D.4) can be rearranged to provide a different perspective:

$$S \overset{\text{def}}{=} Y - (C + G) = I \tag{D.5}$$

The country consumes $(C + G)$ of its produced goods $Y$ as consumer goods within the period. The remaining quantity $S = Y - (C - G)$ can be carried over to future periods and is called savings. The content of savings $S$ ultimately consists of capital goods produced by firms as investment $I$ to increase future production. Thus, Equation (D.5) shows that "current-period savings $S$ are allocated to investment $I$ for future periods."

We simplify savings $S$ as follows. First, simplifying that a fixed proportion $c$ of goods produced by the country is demanded as consumer goods:

$$(C + G) = cY \tag{D.6}$$

Therefore:

$$S = (1 - c)Y = sY \tag{D.7}$$

In other words, of the production quantity $Y$, *a fixed proportion determined by the savings rate* $s \overset{def}{=} (1 - c)$ *is allocated to savings.*

Figure E.5: Household decision prompt $\mathcal{D}^H$.

```
1. Willingness to work
   - Working more will increase your income, and hence increased
     savings, which will increase your future happiness.
   - At the same time, working more will reduce your available leisure
     time, harming your current happiness.
   - json_key='willingness_to_work': A value between 0 and 1
     representing your willingness or propensity to work.

2. Consumption of luxury goods
   - More consumptions will increase your happiness. Less consumptions
     will increase your savings, and hence the future happiness.
   - Consider factors such as your income, past consumption and the
     trend of goods' price, if possible quantitatively.
   - You should calculate the consumption in money explicitly, i.e.,
     doing price times number of goods. Then, compare it to your
     income. While you might occasionally spend two or three times
     your income, going as high as five times or more is clearly
     excessive and financially unsustainable.
   - Provide your decisions based on the amount you would "like" to
     purchase, while you may not be able to purchase the exact amount
     of goods you desire due to stock limitations.
   - json_key='consumer_goods_demand': A value indicating the number of
     luxury goods you plan to buy. Note that this is NOT the
     consumption in dollar but the number of the goods (integer). The
     consumption in dollar will be calculated as this number times
     the current price of luxury goods.
```

# E   LLM AGENTS

## E.1   PERSONA PROMPT

We used random personas (name, age, occupation, residence) of (Li et al., 2024a). A typical persona prompt is like as follows:

> You are Kylie Crawford, a 47-year-old individual living in San Jose, California, working as a high school teacher.

## E.2   DECISION PROMPT

Figures E.5 and E.6 shows the decision prompts listing all decision items. Note that for the households' "willingness to work" ($\in [0,1]$) decision is converted to working hours by $l_t^{(h)} = 168 \times$ (willingness to work).

Figure E.6: Firm decision prompt $\mathcal{D}^F$.

```
1. Consumer Goods Pricing Strategy:
   1. Determine how to adjust the price of your consumer goods.
   2. Note that you are specifying a monthly change rate, typically
      around +-1%.
   3. json_key='consumer_goods_price_change_rate': Rate of change in
      consumer goods price (e.g., 0.1 means 10% increase, -0.1 means
      10% decrease)
2. Capital Goods Pricing Strategy:
   1. Determine how to adjust the price of your capital goods.
   2. Note that you are specifying a monthly change rate, typically
      around +-1%.
   3. json_key='capital_goods_price_change_rate': Rate of change in
      capital goods price (e.g., 0.1 means 10% increase, -0.1 means
      10% decrease)
3. Wage Adjustment Strategy:
   1. Decide how to adjust the wage.
   2. You shuold consider the trend of your total goods sales, while
      incrasing sales allows you to increase wages, decreasing sales
      may require you to decrease wages or at least not increase them.
   3. Note that you are specifying a monthly change rate, typically
      around +-1%.
   4. json_key='wage_change_rate': Rate of change in wages (e.g., 0.1
      means 10% increase, -0.1 means 10% decrease)
4. Investment on New Capital Equipments:
   1. Decide how much to invest from your previous month's profits for
      buying new capital equipments (office equipment, factories,
      etc.), which will increase future production capacity.
   2. The value should be consistent with the ones set by typical
      companies.
   4. json_key='investment_ratio': "Ratio of investment in new capital
      equipment relative to your profit (e.g., 0.1 means 10%. Note:
      Investment is not allowed if profit is negative)."
5. Labor Force Allocation between Goods Production Sector and R&D
    Sector:
   1. When making changes, consider the balance between the current and
      future production, as follows. Production sector directly
      produce goods (both consumer goods and capital goods), expanding
      the current production. While R&D sector doesn't directly
      produce any goods, it increases expand production with
      innovation over the long term.
   2. The value should be consistent with the ones set by typical
      companies.
   4. json_key='R_D_labor_ratio': Proportion of labor force allocated
      to R&D (between 0 and 1. Note that this is **NOT the change rate
      from the previous month**, but the actual absolute ratio)
6. Production Allocation between Consumer Goods and Capital Goods in
    Your Manufacturing Operations:
   1. You can change this allocation depending on which goods will be
      more profitable in the future, considering, for example, the
      demand-production gap or price trend of each goods.
   2. As the total production power is fixed, 5% more allocation to
      capital goods production increases exactly 5% of your capital
      goods production and decreases exactly 5% of your consumer goods
      production.
   3. json_key='capital_goods_production_ratio': Proportion of capital
      goods in total goods in your manufacturing operations (between 0
      and 1. Note that this is **NOT the change rate from the previous
      month**, but the actual absolute ratio)
```

### E.3 ECONOMIC TREND PROMPT

Figure E.7 shows the economic trend prompt, which LLM agents reference during decision-making. Table E.3 lists all the economic variables included in the economic trend prompts.

As shown, the LLM agents mainly referred to the current $t$ and immediate past $t-1$ variables. For the results when LLM agents refer economic variables further back in time, see Section H.7.

Figure E.7: Economic trend prompts. For full list of economic variables shown in these prompts, refer to Table E.3.

(a) Household economic trend prompt $\mathcal{E}_t^{(h)}$.

```
# Economic Variables

(...)
## Job and income
* Monthly Income (if worked full-time)
    - 2 months ago : 3685.58
    - Last month : 3648.72
(...)

## Consumer Goods
* Prices:
    - 2 months ago : 101.52
    - Last month : 101.65
* Your demand in number of goods:
    - Last month : 32
* Your consumption in dollar:
    - Last month : 3248.70
(...)
```

(b) Firm economic trend prompt $\mathcal{E}_t^{(f)}$.

```
# Economic Variables

## Supply and Demand
(...)
* The production of consumer goods:
    - Last month : 28
* The demand of consumer goods:
    - Last month : 159
* The quantity of consumer goods sold (may not match the demand due to
   the stock limitation):
    - Last month : 159

(...)
* The production of capital goods:
    - Last month : 114
(...)

## Production Strategy
* The proportion of capital goods in total production
    ("capital_goods_production_ratio"):
    - Last month : 0.80
* The proportion of labor force allocated to R&D between 0 and 1 (i.e.,
    "R_D_labor_ratio"):
    - Last month : 0.05

## Prices
* Your company's price for consumer goods in dollars:
    - 2 months ago : 110.16
    - Last month : 111.26
* Your company's price for capital goods in dollars:
    - 2 months ago : 81.86
    - Last month : 81.04

## Financials
(...)
* The total personnel cost in dollars (wages multiplied by labor force):
    - 2 months ago : 11307.15
    - Last month : 11448.60
* The profit:
    - Last month : 9904.56
(...)
```

Table E.3: Economic variables included in the economic trend prompts, which LLM agents reference during decision-making. As shown, the LLM agents mainly referred to the current $t$ and immediate past $t-1$ variables. For the results when LLM agents refer economic variables further back in time, see Section H.7.

(a) Variables included in households' prompt $\mathcal{E}_t^{(h)}$.

| Labor and income | Consumer goods consumption | Finance |
|---|---|---|
| labor hour: $l_{t-1}^{(h)}$
income: $i_{t-1}^{(h)}, i_{t-2}^{(h)}$ | price: $p_t^C, p_{t-1}^C$
consumption (quantity): $d_{t-1}^{C(h) \to (\sigma_t^H)}$
consumption (dollars): $c_{t-1}^{(h)}$ | Current savings: $b_t^{(h)}$
Current interest rate. |

(b) Variables included in firms' prompt $\mathcal{E}_t^{(f)}$.

| Consumer goods | Capital goods | Total goods | Labor | Finance |
|---|---|---|---|---|
| – | production ratio: $r_{t-1}^{K(f)}$ | – | total labor hours $L_t^{(f)}$ | employee wage: $\text{avg}_{h \in E^{(f)}} w_t^{(h)}$ |
| production: $Y_{t-1}^{C(f)}$ | production: $Y_t^{K(f)}$ | production $Y_t^{(f)}$ | R&D ratio: $r_{t-1}^{RD(f)}$ | personnel cost: $C_{t-1}^{(f)}$ |
| price: $p_{t-1}^{C(f)}$ | price: $p_{t-1}^{K(f)}$ | – | – | capital investment: $\overline{M}_{t-1}^{I(f)}$ |
| demand: $D_{t-1}^{C(f)}$ | demand: $D_{t-1}^{K(f)}$ | demand: $D_{t-1}^{C(f)} + D_{t-1}^{K(f)}$ | – | profit: $P_{t-1}^{(f)}$ |
| sold: $n_{t-1}^{C(f)}$ | sold: $n_{t-1}^{K(f)}$ | sold: $n_{t-1}^{C(f)} + n_{t-1}^{K(f)}$ | – | retained earnings: $R_t^{(f)}$ |
| sales (dollars): $S_{t-1}^{C(f)}$ | sales (dollars): $S_{t-1}^{K(f)}$ | sales (dollars): $S_{t-1}^{(f)}$ | – | – |
| inventory: $V_t^{C(f)}$ | inventory: $V_t^{K(f)}$ | inventory: $V_t^{C(f)} + V_t^{K(f)}$ | – | – |

Figure E.8: A sample of LLM outputs from firm agents.

```
I'll analyze each decision point based on the provided information:

1. Consumer Goods Pricing Strategy:
  - Current demand (30) is double the production (15)
  - High inventory level (45) suggests we can meet demand
  - Given high demand and sufficient inventory, we can increase price
      moderately
  - Suggest 1% price increase to capitalize on strong demand while
      staying competitive
Thought: increase by 1% (0.01)

2. Capital Goods Pricing Strategy:
  - Similar situation to consumer goods (demand 30, production 15)
  - Healthy inventory levels (45)
  - Market conditions similar to consumer goods
  - Maintain pricing parity with consumer goods
Thought: increase by 1% (0.01)

3. Wage Adjustment Strategy:
  - Strong sales (5700.70)
  - Positive profit (9501.16)
  - High demand for both product categories
  - Can afford modest wage increase to maintain worker satisfaction
Thought: increase by 0.5% (0.005)

4. Investment on New Capital Equipments:
  - Strong profit (9501.16)
  - High demand-to-production gap suggests need for capacity expansion
  - Current investment ratio is about 30% (2850.35/9501.16)
  - Maintain similar investment level to support growth
Thought: set to 0.3 (30%)

5. Labor Force Allocation between Production and R&D:
  - Current R&D ratio is very low (0.05)
  - High demand suggests need for both current production and future
      innovation
  - Slightly increase R&D while maintaining strong production
Thought: set to 0.07 (7%)

6. Production Allocation:
  - Equal demand for both consumer and capital goods
  - Current 50-50 split matches demand pattern
  - No clear advantage to changing the mix
Thought: maintain at 0.5 (50%)

```json
{
    "consumer_goods_price_change_rate": 0.01,
    "capital_goods_price_change_rate": 0.01,
    "wage_change_rate": 0.005,
    "investment_ratio": 0.3,
    "R_D_labor_ratio": 0.07,
    "capital_goods_production_ratio": 0.5
}
```
```

## E.4 LLM OUTPUT

Figure E.8 shows a sample of LLM outputs.

# F  ECONGROWTHAGENT

Below are the remaining equations of EconGrowthAgent.

## F.1  HOUSEHOLDS

### F.1.1  CONSUMPTION AND SAVINGS

Households deposit the remainder of their income after consumption in the bank.

$$\text{monthly saving } m_t^{(h)} = i_t^{(h)} - c_t^{(h)} \tag{F.1}$$

$$\text{balance } b_{t+1}^{(h)} = b_t^{(h)} + m_t^{(h)} \tag{F.2}$$

## F.2  FIRMS

### F.2.1  TECHNOLOGY LEVEL IMPROVEMENT THROUGH R&D

The R&D department improves the firm's technology level $A_t^{(f)}$:

$$A_{t+1}^{(f)} = A_t^{(f)} + \underbrace{\gamma \frac{1}{\tau^{RD}} \sum_{t'=1}^{\tau^{RD}} L_{t-t'}^{RD(f)} A_{t-t'}^{(f),\phi}}_{\Delta A_t^{(f)}} \tag{F.3}$$

The growth rate of technology level depends on the labor devoted to R&D $L_t^{RD}$. However, it takes time for R&D to yield results. Here, R&D results are assumed to be produced by an accumulation over $\tau^{RD}$ months.

### F.2.2  GOODS SALES

Goods inventory changes with sales:

$$\text{consumer goods inventory } V_{t+1}^{C(f)} = V_t^{C(f)} + (Y_t^{C(f)} - D_t^{C(f)}) \tag{F.4}$$

$$\text{capital goods inventory } V_{t+1}^{K(f)} = V_t^{K(f)} + (Y_t^{K(f)} - D_t^{K(f)}) \tag{F.5}$$

## F.3  BANK

The bank is responsible for investing deposits. Specifically, it invests monthly savings from households $\sum_{h \in [1,N^H]} m_t^{(h)}$ in firms' stock purchases. For simplicity, the stock portfolio is allocated proportionally to each firm's number of employees: that is, stocks worth $F_t^{(f)} = \frac{|E^{(f)}|}{N^H} \times \sum_{h \in [1,N^H]} m_t^{(h)}$ are purchased from firm $f$.

# G    Experimental Setup

## G.1    General Setting

▶ **Simulation Settings:** We chose $N^H + N^F = 100$ agents to balance economic realism with computational costs. We tested configurations with 10, 30, 100, and 300 agents. With 10 and 30 agents, individual agent characteristics dominated the system, making macro-level phenomena unstable. With 100 and 300 agents, macro-level phenomena remained stable. However, 300 agents incurred approximately \$200 in API costs per simulation run. Therefore, we selected 100 agents as the optimal balance.

Our $N^H : N^F = 90 : 10$ split lies comfortably within the empirically observed band. OECD data show household-to-firm ratios ranging from roughly 8:1 to 25:1 across member countries. We chose the specific 9:1 ratio for computational convenience, i.e., it partitions 100 agents into integers (90 households, 10 firms).

▶ **Economic Constants:** We used real-world measurements where available. For instance, we sampled initial household wages from the 2024 U.S. wage distribution (U.S. Labor Statistics, 2024), set $\alpha$ in the Cobb–Douglas function eq. (2) to 0.3 (Solow, 1956a), used a three-year R&D delay $\tau^{RD}$ in Equation (F.3) (Hall, 2002), and set the technology self-growth parameter $\phi$ in eq. (5) to 1.0 (Romer, 1990).

For other constants in eq. (2), we adjusted parameters so that the capital and technology factors each achive reasonable growth, i.e., about 5% per year. See the source code for further details.

▶ **LLM Agents:** We modeled each household as a single individual for conceptual clarity and to keep the first-version framework simple. Extending to family units could capture phenomena such as intra-family consumption discussions, but this would require implementing inter-agent communication mechanisms, which we reserve for future work.

▶ **Employee Assignment:** Each household uniformly selects one firm to work for.

## G.2    Setting for Business Performance Analysis in Section H.6

We used $N^H = 90$ household agents and $N^F = 9$ firm agents. With three types of LLMs (GPT-4.1, GPT-4.1-nano, and GPT-3.5-Turbo), each LLM was represented by three firm agents. Results show the average performance of these three firms for each LLM type. Additionally, we repeated the experiment 10 times with different random seeds and present the averaged results.

For a fair comparison, each firm was assigned the same number of employees (10) with identical personas, and the funding $F_t^{(f)}$ was kept constant across firms.

## G.3    Setting for Decision-Making Pattern Analysis in Section 5.3

The setting of the simulation is the same as Section G.2.

See Section H.5 for the regression setting.

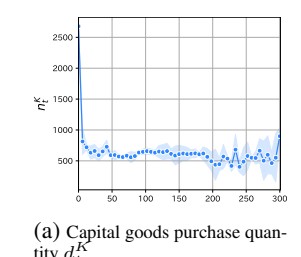

(a) Capital goods purchase quantity $d_t^K$

Figure H.9: Simulation results over 25 years ($t \in [1, 300]$, horizontal axis) using Claude-3.5-Sonnet agents (other economic variables).

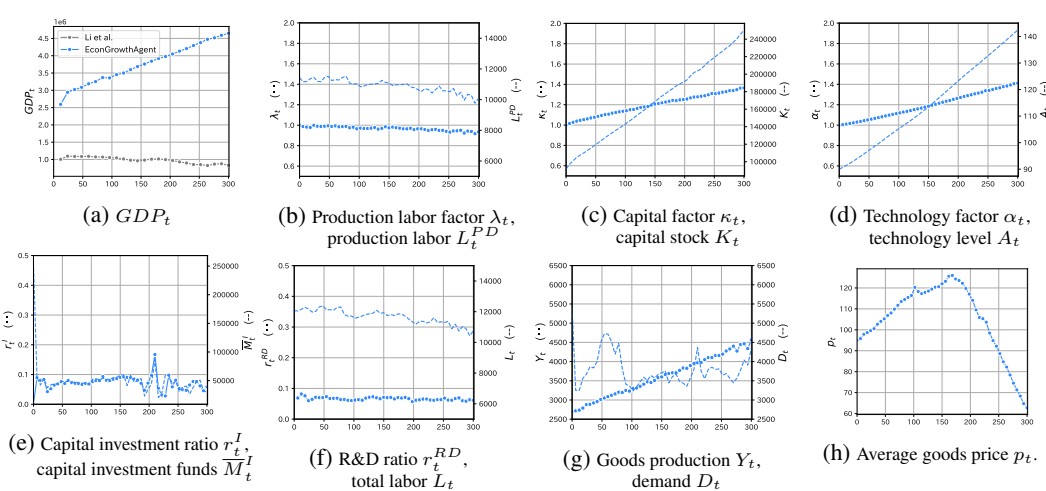

(a) $GDP_t$

(b) Production labor factor $\lambda_t$, production labor $L_t^{PD}$

(c) Capital factor $\kappa_t$, capital stock $K_t$

(d) Technology factor $\alpha_t$, technology level $A_t$

(e) Capital investment ratio $r_t^I$, capital investment funds $\overline{M}_t^I$

(f) R&D ratio $r_t^{RD}$, total labor $L_t$

(g) Goods production $Y_t$, demand $D_t$

(h) Average goods price $p_t$.

Figure H.10: Simulation results over 25 years ($t \in [1, 300]$, horizontal axis) using *GPT-4.1* agents.

# H    OTHER EXPERIMENTAL RESULTS

## H.1    ECONOMIC GROWTH

Figure H.9 shows additional economic variables for the "economic growth" simulation discussed in Section 5.1.

Figure H.10 shows the same simulation using GPT-4.1 as LLM agents. The results are consistent with those using Claude-3.5-SOnnet, demonstrating robustness across different backbone LLMs.

The price $p_t$ average across both consumer and capital goods is defined as follows:

$$p_t = \sum_f (p_t^{C(f)} Y_t^{C(f)} + p_t^{K(f)} Y_t^{K(f)}) / \sum_f (Y_t^{C(f)} + Y_t^{K(f)}) \tag{H.1}$$

GDP uses the base year price $p_{t=1}$ to exclude the effects of price fluctuations. In contrast, nominal GDP$_t = Y_t \times p_t$ represents the "current value" of production, but even with identical production quantity $Y_t$, it fluctuates with price $p_t$, making it unsuitable as an indicator of a country's prosperity.

The decomposition of GDP is defined as follows:

$$\lambda_t = \sum_{f \in [1, N^F]} (Y_t^{(f)} \times \lambda_t^{(f)}) / \sum_{f \in [1, N^F]} Y_t^{(f)} \tag{H.2}$$

Weighting by goods production $Y_t^{(f)}$ considers the different impact on GDP by firm.

We reimplemented Li et al. (2023) by EconGrowthAgent with $\alpha = 0, \gamma = 0$.

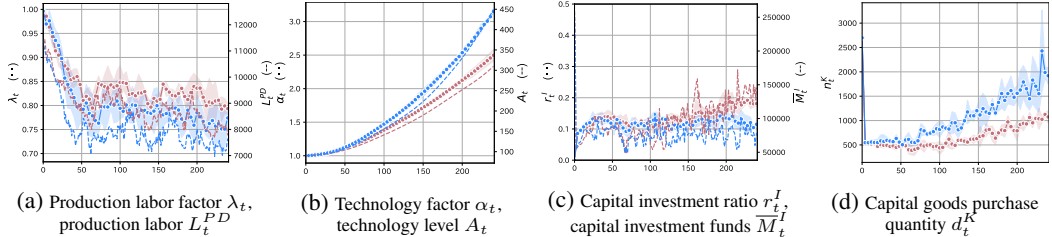

(a) Production labor factor $\lambda_t$, production labor $L_t^{PD}$

(b) Technology factor $\alpha_t$, technology level $A_t$

(c) Capital investment ratio $r_t^I$, capital investment funds $\overline{M}_t^I$

(d) Capital goods purchase quantity $d_t^K$

Figure H.11: Comparison between a frugal country and a spendthrift country, using Claude-3.5-Haiku agents (other economic variables).

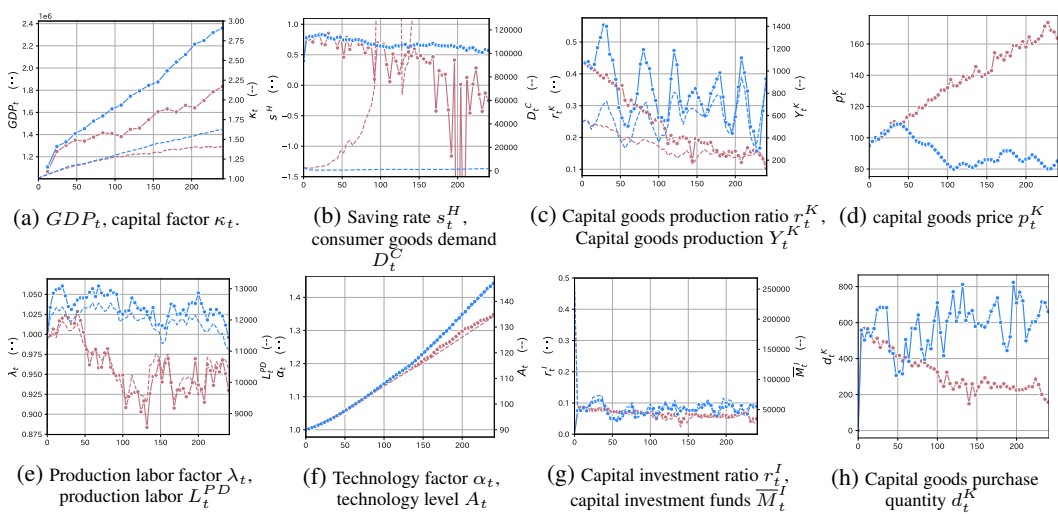

(a) $GDP_t$, capital factor $\kappa_t$.

(b) Saving rate $s_t^H$, consumer goods demand $D_t^C$

(c) Capital goods production ratio $r_t^K$, Capital goods production $Y_t^K$

(d) capital goods price $p_t^K$

(e) Production labor factor $\lambda_t$, production labor $L_t^{PD}$

(f) Technology factor $\alpha_t$, technology level $A_t$

(g) Capital investment ratio $r_t^I$, capital investment funds $\overline{M}_t^I$

(h) Capital goods purchase quantity $d_t^K$

Figure H.12: Comparison between a frugal country and a spendthrift country, using *GPT-4.1* agents.

## H.2 RELATED KEY PHENOMENON: HIGH GROWTH IN FRUGAL COUNTRIES

Figure H.11 shows additional economic variables for the "a frugal nation vs a spendthrift nation" simulation discussed in Section 5.2.

Figure H.12 shows the same simulation using Claude-3.5-Sonnet as LLM agents. The results are consistent with those using GPT-4.1, demonstrating robustness across different backbone LLMs.

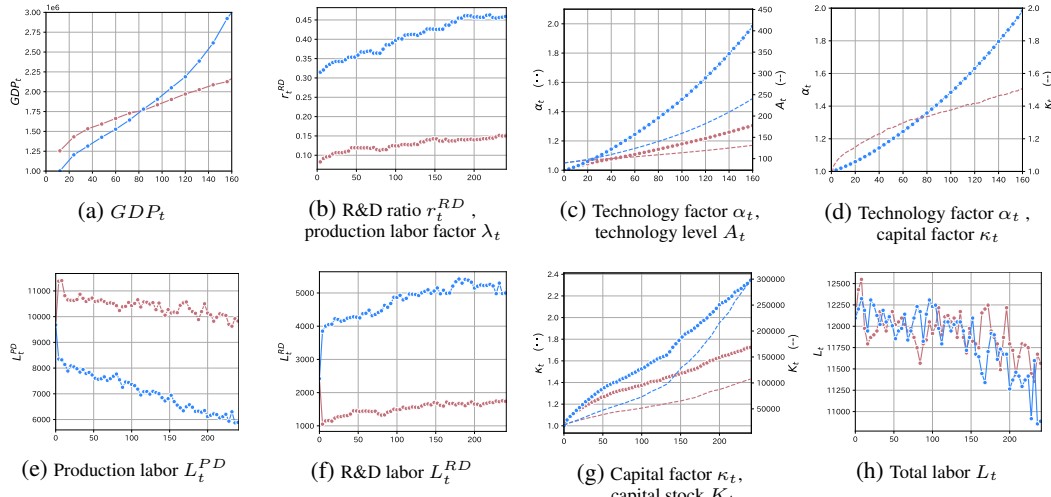

(a) $GDP_t$

(b) R&D ratio $r_t^{RD}$, production labor factor $\lambda_t$

(c) Technology factor $\alpha_t$, technology level $A_t$

(d) Technology factor $\alpha_t$, capital factor $\kappa_t$

(e) Production labor $L_t^{PD}$

(f) R&D labor $L_t^{RD}$

(g) Capital factor $\kappa_t$, capital stock $K_t$

(h) Total labor $L_t$

Figure H.13: Comparison between a technology-innovative nation and a conservative nation. We used Claude-3.5-Haiku as the LLM.

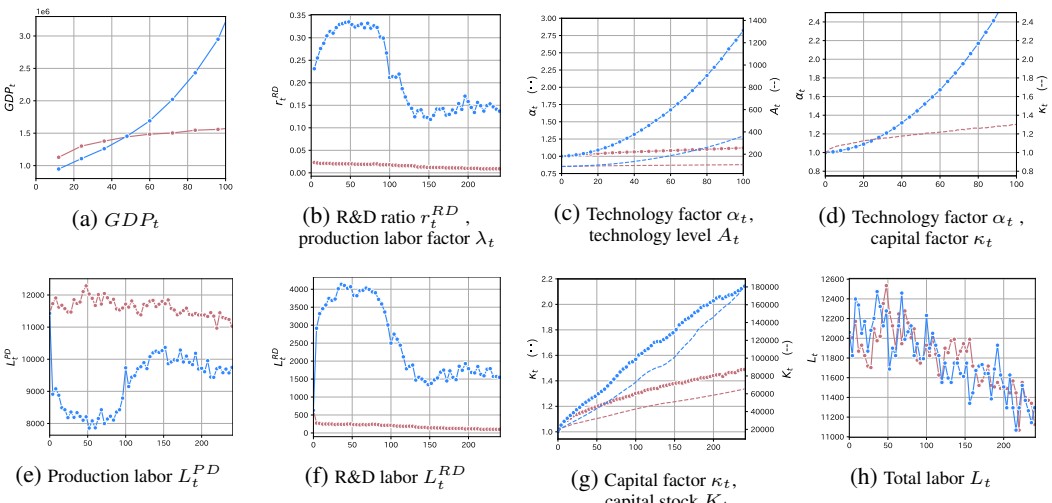

(a) $GDP_t$

(b) R&D ratio $r_t^{RD}$, production labor factor $\lambda_t$

(c) Technology factor $\alpha_t$, technology level $A_t$

(d) Technology factor $\alpha_t$, capital factor $\kappa_t$

(e) Production labor $L_t^{PD}$

(f) R&D labor $L_t^{RD}$

(g) Capital factor $\kappa_t$, capital stock $K_t$

(h) Total labor $L_t$

Figure H.14: Comparison between a technology-innovative nation and a conservative nation, using *GPT-4.1 agents*.

## H.3 RELATED KEY PHENOMENON: SUSTAINABLE GROWTH THROUGH TECHNOLOGICAL INNOVATION

We show that EconGrowthAgent can reproduce and analyze growth-related key phenomena, namely, "technological innovation enables long-term growth" (Romer, 1986a).

The results are consistent with those using Claude-3.5-Sonnet (fig. H.13) and GPT-4.1 (fig. H.14), demonstrating robustness across different backbone LLMs.

▶ **Reproduction of the Macro-level Phenomenon:**

Figure H.13 compares economic growth between a "technology-innovative nation" and "conservative nation". Technology-innovative nation firms were given the persona prompt: "In this country, technological innovation is worshipped and pursued. Citizens celebrate breakthroughs in science and technology." For the conservative nation, the prompt was: "Innovation is not highly valued in this country, and people engaged in activities such as R&D are often considered to be 'playing around.'"

As shown in fig. H.13a, while the technology-innovative nation had lower GDP in the initial stages, it eventually surpassed the conservative nation.

▶ **Micro-Level Analysis**

Figure H.13b (left) indicates that the technology-innovative nation maintained a high R&D ratio $r_t^{RD}$ eq. (8), whereas the conservative nation maintained a low ratio. Decision-making in each country was as follows:

> *Given the strong culture of innovation and the government's support for R&D, it would be beneficial for the company to allocate a significant portion of its labor force to the R&D sector. This would help the company develop new and innovative products, which could lead to increased sales and profitability in the long run.*
>
> *Given the low value placed on innovation in the country, I will allocate a smaller proportion of the labor force to the R&D sector to focus on immediate production needs.*

That is, the technology-innovative nation allocated labor to R&D with a long-term perspective, while the conservative nation allocated labor to production to meet immediate demand.

As a result of this decision, the technology-innovative nation had less production labor eq. (9) (fig. H.13e) and a lower production labor factor $\lambda_t$ eq. (18) (fig. H.13b right). This explains its initially lower GDP.

However, with higher R&D labor eq. (9) (fig. H.13f) in the technology-innovative nation, once R&D results began to materialize (after $t \geq \tau^{RD} = 36$), the technology factor eq. (18) increased substantially (fig. H.13c left), enabling its GDP to surpass the conservative nation.

Conservative the nation allocated minimal labor to R&D (fig. H.13f), resulting in extremely slow growth in the technology factor (fig. H.13c left). Instead, they allocated more labor to goods production (fig. H.13e). While production labor did not grow, thus not leading to GDP growth, greater production labor could produce more capital goods, potentially leading to faster capital stock accumulation. In this sense, the conservative nation' economic growth relied on capital stock accumulation.

Hence, the comparison between technology-innovative and the conservative nation can be viewed as a comparison between "a nation relying on technology level improvement for economic growth" and "a nation relying on capital stock accumulation."

Figure H.13d shows the *technology* factor $\alpha_t$ for the technology-innovative nation and the *capital* factor $\kappa_t$ for the conservative nation. While the technology factor grew continuously, the capital factor diminished (saturated).

The reason technology factor growth does not diminish is due to the property that technological improvement can "stand on the shoulders of giants" (Sections 2.4 and H.3). That is, the larger the available technology and knowledge at a given point, the easier it is to create new technology, making technology level growth less prone to diminishing. Conversely, growth through capital stock accumulation is more prone to diminishing due to limits on the amount of capital a worker can handle (Section H.3).

The fact that *long-term sustainable economic growth is only possible through technological innovation* (Romer, 1986a) is an essential finding in economics.

▶ **Mathematical Explanation:**

We show a mathematical explanation of diminishing growth by capital accumulation and sustainable growth by technological innovation.

The reason technology factor growth does not diminish is due to the property that technological improvement can "stand on the shoulders of giants" (Section 2.4). Rearranging eq. (5) and roughly solving it by extracting only the dependence on technology level $A_t$:

$$A_{t+1} - A_t = \gamma A_t \tag{H.3}$$

(where $\phi \sim 1$, which approximately holds for many economies). This solution clearly increases exponentially with time (in continuous time, the equation becomes $\frac{dA}{dt} = \gamma A$, with solution $A_t = e^{\gamma t}$). Although exponential growth may seem counterintuitive, it aligns with real-world observations. For instance, advanced countries maintain roughly constant GDP growth rates; if the economy itself is growing, a steady *rate* implies that its *absolute growth* (the increment in GDP) increases exponentially.

Conversely, extracting only the main terms from the capital stock growth equation eq. (4):

$$K_{t+1} - K_t = \frac{sF^0}{(K^0)^\alpha} \times K^\alpha \tag{H.4}$$

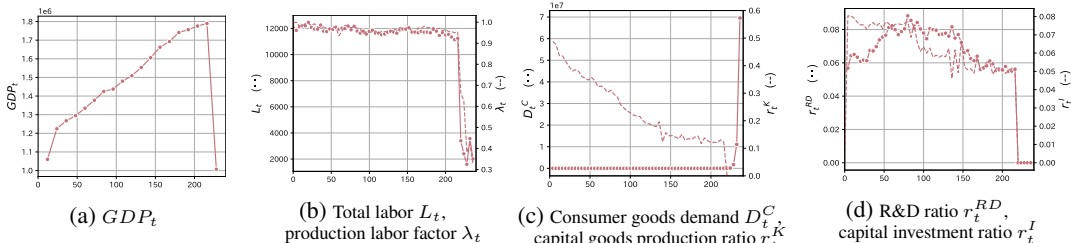

(a) $GDP_t$

(b) Total labor $L_t$,
production labor factor $\lambda_t$

(c) Consumer goods demand $D_t^C$,
capital goods production ratio $r_t^K$

(d) R&D ratio $r_t^{RD}$,
capital investment ratio $r_t^I$

Figure H.15: Counterfactual simulation of "forecast of an approaching civilization-ending asteroid", using *GPT-4.1* agents.

$\alpha < 1$ causes growth to saturate as capital stock $K_t$ increases, reflecting the diminishing returns property of the production function eq. (D.3). As there is a limit to the amount of materials and equipment a worker can handle, economic growth through capital stock accumulation also has limits.

## H.4 COUNTERFACTUAL SIMULATION WITH ECONGROWTHAGENT

Figure H.10 shows the counterfactual simulation discussed in Section 6 using Claude-3.5-Sonnet as LLM agents. The results are consistent with those using GPT-4.1, demonstrating robustness across different backbone LLMs.

Table H.4: All the regression coefficients of the analysis are discussed in Section H.5. Variables: $S_t$="sales", $Y_t$="goods production", $D_t$="demand", $g_t = (D_t - Y_t)/\max(D_t, Y_t)$="demand-supply gap", $i_t$="price change ra", $C_t$="personnel cost", $F_t$="bank financing", $\hat{M}_t$="fund after wage payments", $s_t^F$="=savings (i.e., retained earnings minus debt from bank financing)", $P_t$="profit", $t$="trend term". Variables with hat $\hat{X}_t$ represent rate of change: $\hat{X}_t = (X_t - X_{t-1})/X_{t-1}$.

(a) GPT-4.1.

| | $\hat{S}_{t-1}$ | $\hat{D}_{t-1}$ | $D_{t-1}^C$ | $g_{t-1}^C$ | $i_{t-1}^C$ | $D_{t-1}^K$ | $g_{t-1}^K$ |
|---|---|---|---|---|---|---|---|
| consumer goods price change rate $i_t^C$ | $0.21_{\pm0.025}$ | $-0.0034_{\pm0.014}$ | $-0.099_{\pm0.013}$ | $\mathbf{0.73}_{\pm0.012}$ | $0.2_{\pm0.01}$ | $-0.065_{\pm0.014}$ | $-0.035_{\pm0.01}$ |
| capital goods price change rate $i_t^K$ | $0.028_{\pm0.024}$ | $0.0094_{\pm0.014}$ | $0.09_{\pm0.013}$ | $0.0055_{\pm0.012}$ | $0.024_{\pm0.0098}$ | $-0.05_{\pm0.014}$ | $\mathbf{0.69}_{\pm0.01}$ |
| wage change rate $i_t^w$ | $\mathbf{0.58}_{\pm0.038}$ | $-0.12_{\pm0.022}$ | $0.044_{\pm0.02}$ | $0.24_{\pm0.019}$ | $-0.076_{\pm0.016}$ | $-0.038_{\pm0.022}$ | $0.19_{\pm0.016}$ |
| capital goods production ratio $r_t^K$ | $-0.19_{\pm0.035}$ | $0.084_{\pm0.02}$ | $\mathbf{-0.33}_{\pm0.018}$ | $0.052_{\pm0.017}$ | $-0.26_{\pm0.014}$ | $\mathbf{0.49}_{\pm0.02}$ | $0.048_{\pm0.014}$ |
| R&D ratio $r_t^{RD}$ | $0.21_{\pm0.026}$ | $-0.063_{\pm0.015}$ | $0.1_{\pm0.014}$ | $-0.13_{\pm0.013}$ | $-0.13_{\pm0.011}$ | $0.16_{\pm0.015}$ | $-0.05_{\pm0.011}$ |
| capital investment ratio $r_t^I$ | $0.01_{\pm0.038}$ | $-0.014_{\pm0.022}$ | $0.024_{\pm0.02}$ | $0.16_{\pm0.018}$ | $-0.059_{\pm0.015}$ | $0.1_{\pm0.021}$ | $0.19_{\pm0.016}$ |

| | $i_{t-1}^K$ | $\hat{C}_{t-1}$ | $\hat{F}_{t-1}$ | $\overline{\hat{M}}_{t-1}$ | $\hat{s}_{t-1}^F$ | $P_{t-1}$ | $t$ |
|---|---|---|---|---|---|---|---|
| consumer goods price change rate $i_t^C$ | $-0.006_{\pm0.0092}$ | $0.015_{\pm0.0073}$ | $0.0015_{\pm0.028}$ | $-0.035_{\pm0.028}$ | $0.0043_{\pm0.022}$ | $-0.016_{\pm0.009}$ | $0.00013_{\pm0.0072}$ |
| capital goods price change rate $i_t^K$ | $0.21_{\pm0.0089}$ | $0.014_{\pm0.0071}$ | $-0.0021_{\pm0.027}$ | $-0.014_{\pm0.027}$ | $0.023_{\pm0.021}$ | $-0.023_{\pm0.0087}$ | $0.0027_{\pm0.007}$ |
| wage change rate $i_t^w$ | $-0.062_{\pm0.014}$ | $0.12_{\pm0.011}$ | $0.036_{\pm0.043}$ | $-0.023_{\pm0.043}$ | $-0.082_{\pm0.034}$ | $0.22_{\pm0.014}$ | $-0.026_{\pm0.011}$ |
| capital goods production ratio $r_t^K$ | $0.24_{\pm0.013}$ | $-0.0063_{\pm0.01}$ | $0.079_{\pm0.039}$ | $0.057_{\pm0.039}$ | $0.046_{\pm0.031}$ | $-0.12_{\pm0.013}$ | $-0.086_{\pm0.01}$ |
| R&D ratio $r_t^{RD}$ | $-0.0028_{\pm0.0097}$ | $0.02_{\pm0.0077}$ | $0.03_{\pm0.029}$ | $0.019_{\pm0.029}$ | $0.01_{\pm0.023}$ | $0.27_{\pm0.0095}$ | $0.031_{\pm0.0075}$ |
| capital investment ratio $r_t^I$ | $0.076_{\pm0.014}$ | $0.048_{\pm0.011}$ | $-0.073_{\pm0.043}$ | $-0.044_{\pm0.042}$ | $-0.11_{\pm0.033}$ | $0.16_{\pm0.014}$ | $0.037_{\pm0.011}$ |

(b) GPT-4.1-nano.

| | $\hat{S}_{t-1}$ | $\hat{D}_{t-1}$ | $D_{t-1}^C$ | $g_{t-1}^C$ | $i_{t-1}^C$ | $D_{t-1}^K$ | $g_{t-1}^K$ |
|---|---|---|---|---|---|---|---|
| consumer goods price change rate $i_t^C$ | $-0.018_{\pm0.076}$ | $-0.018_{\pm0.076}$ | $1.2\text{e-}16_{\pm2e-16}$ | $3.7\text{e-}16_{\pm3.3e-16}$ | $-0.12_{\pm0.14}$ | $-0.85_{\pm0.7}$ | $0.57_{\pm0.58}$ |
| capital goods price change rate $i_t^K$ | $-0.087_{\pm0.073}$ | $-0.087_{\pm0.073}$ | $4.3\text{e-}17_{\pm1.9e-16}$ | $2.8\text{e-}17_{\pm3.1e-16}$ | $-0.25_{\pm0.13}$ | $0.2_{\pm0.67}$ | $0.27_{\pm0.56}$ |
| wage change rate $i_t^w$ | $0.046_{\pm0.07}$ | $0.046_{\pm0.071}$ | $-1\text{e-}16_{\pm1.9e-16}$ | $-6.3\text{e-}17_{\pm3.1e-16}$ | $-0.09_{\pm0.13}$ | $0.56_{\pm0.65}$ | $-0.099_{\pm0.54}$ |
| capital goods production ratio $r_t^K$ | $-0.026_{\pm0.066}$ | $-0.026_{\pm0.066}$ | $-1.5\text{e-}16_{\pm1.8e-16}$ | $9.8\text{e-}17_{\pm2.9e-16}$ | $-0.19_{\pm0.12}$ | $0.024_{\pm0.61}$ | $0.12_{\pm0.5}$ |
| R&D ratio $r_t^{RD}$ | $-0.059_{\pm0.051}$ | $-0.059_{\pm0.051}$ | $1.8\text{e-}17_{\pm1.4e-16}$ | $2.2\text{e-}16_{\pm2.2e-16}$ | $0.0032_{\pm0.094}$ | $-0.27_{\pm0.47}$ | $0.32_{\pm0.39}$ |
| capital investment ratio $r_t^I$ | $-0.054_{\pm0.065}$ | $-0.054_{\pm0.065}$ | $-1\text{e-}16_{\pm1.7e-16}$ | $-2.2\text{e-}16_{\pm2.8e-16}$ | $-0.17_{\pm0.12}$ | $0.49_{\pm0.6}$ | $-0.32_{\pm0.5}$ |

| | $i_{t-1}^K$ | $\hat{C}_{t-1}$ | $\hat{F}_{t-1}$ | $\overline{\hat{M}}_{t-1}$ | $\hat{s}_{t-1}^F$ | $P_{t-1}$ | $t$ |
|---|---|---|---|---|---|---|---|
| consumer goods price change rate $i_t^C$ | $-0.046_{\pm0.14}$ | $-0.15_{\pm0.12}$ | $0.1_{\pm0.53}$ | $0.12_{\pm0.23}$ | $-0.13_{\pm0.15}$ | $0.19_{\pm0.13}$ | $-0.021_{\pm0.11}$ |
| capital goods price change rate $i_t^K$ | $-0.058_{\pm0.13}$ | $-0.06_{\pm0.11}$ | $0.17_{\pm0.5}$ | $-0.053_{\pm0.22}$ | $0.087_{\pm0.14}$ | $0.17_{\pm0.13}$ | $0.019_{\pm0.1}$ |
| wage change rate $i_t^w$ | $-0.18_{\pm0.13}$ | $0.17_{\pm0.11}$ | $-0.34_{\pm0.49}$ | $-0.12_{\pm0.22}$ | $0.13_{\pm0.14}$ | $0.1_{\pm0.12}$ | $-0.028_{\pm0.098}$ |
| capital goods production ratio $r_t^K$ | $0.11_{\pm0.12}$ | $0.098_{\pm0.1}$ | $-0.69_{\pm0.46}$ | $-0.17_{\pm0.2}$ | $-0.13_{\pm0.13}$ | $-0.18_{\pm0.11}$ | $\mathbf{-0.47}_{\pm0.092}$ |
| R&D ratio $r_t^{RD}$ | $0.017_{\pm0.095}$ | $-0.026_{\pm0.078}$ | $0.12_{\pm0.35}$ | $0.12_{\pm0.16}$ | $0.096_{\pm0.1}$ | $-0.046_{\pm0.088}$ | $\mathbf{0.77}_{\pm0.071}$ |
| capital investment ratio $r_t^I$ | $0.086_{\pm0.12}$ | $0.2_{\pm0.1}$ | $-0.25_{\pm0.45}$ | $-0.25_{\pm0.2}$ | $0.14_{\pm0.13}$ | $-0.045_{\pm0.11}$ | $-0.1_{\pm0.091}$ |

(c) GPT-3.5-Turbo.

| | $\hat{S}_{t-1}$ | $\hat{D}_{t-1}$ | $D_{t-1}^C$ | $g_{t-1}^C$ | $i_{t-1}^C$ | $D_{t-1}^K$ | $g_{t-1}^K$ |
|---|---|---|---|---|---|---|---|
| consumer goods price change rate $i_t^C$ | $-1.3_{\pm1.2}$ | $1.2_{\pm0.96}$ | $20.0_{\pm17.0}$ | $20.0_{\pm17.0}$ | $\mathbf{0.43}_{\pm0.098}$ | $-0.59_{\pm0.62}$ | $-28.0_{\pm24.0}$ |
| capital goods price change rate $i_t^K$ | $-0.98_{\pm1.2}$ | $1.7_{\pm0.92}$ | $22.0_{\pm16.0}$ | $22.0_{\pm16.0}$ | $0.2_{\pm0.094}$ | $\mathbf{-1.4}_{\pm0.59}$ | $-32.0_{\pm23.0}$ |
| wage change rate $i_t^w$ | $-1.2_{\pm1.2}$ | $0.17_{\pm0.93}$ | $11.0_{\pm16.0}$ | $11.0_{\pm16.0}$ | $\mathbf{0.32}_{\pm0.094}$ | $-0.14_{\pm0.6}$ | $-15.0_{\pm23.0}$ |
| capital goods production ratio $r_t^K$ | $-0.31_{\pm1.2}$ | $0.61_{\pm0.9}$ | $7.4_{\pm16.0}$ | $7.4_{\pm16.0}$ | $-0.014_{\pm0.091}$ | $-0.42_{\pm0.58}$ | $-11.0_{\pm22.0}$ |
| R&D ratio $r_t^{RD}$ | $0.37_{\pm0.96}$ | $0.35_{\pm0.75}$ | $-0.064_{\pm13.0}$ | $-0.064_{\pm13.0}$ | $-0.095_{\pm0.076}$ | $-0.092_{\pm0.48}$ | $-0.11_{\pm18.0}$ |
| capital investment ratio $r_t^I$ | $-1.1_{\pm1.2}$ | $-0.12_{\pm0.97}$ | $7.3_{\pm17.0}$ | $7.3_{\pm17.0}$ | $0.068_{\pm0.099}$ | $0.55_{\pm0.63}$ | $-8.5_{\pm24.0}$ |

| | $i_{t-1}^K$ | $\hat{C}_{t-1}$ | $\hat{F}_{t-1}$ | $\overline{\hat{M}}_{t-1}$ | $\hat{s}_{t-1}^F$ | $P_{t-1}$ | $t$ |
|---|---|---|---|---|---|---|---|
| consumer goods price change rate $i_t^C$ | $-0.0048_{\pm0.1}$ | $-0.036_{\pm0.1}$ | $0.49_{\pm0.52}$ | $-0.075_{\pm0.13}$ | $-0.43_{\pm0.3}$ | $-0.17_{\pm0.28}$ | $-0.12_{\pm0.099}$ |
| capital goods price change rate $i_t^K$ | $0.26_{\pm0.096}$ | $-0.011_{\pm0.099}$ | $0.95_{\pm0.49}$ | $0.044_{\pm0.13}$ | $0.39_{\pm0.28}$ | $\mathbf{0.74}_{\pm0.26}$ | $-0.055_{\pm0.095}$ |
| wage change rate $i_t^w$ | $-0.086_{\pm0.096}$ | $0.26_{\pm0.1}$ | $0.75_{\pm0.5}$ | $0.18_{\pm0.13}$ | $0.057_{\pm0.29}$ | $-0.072_{\pm0.27}$ | $-0.025_{\pm0.096}$ |
| capital goods production ratio $r_t^K$ | $0.15_{\pm0.093}$ | $0.1_{\pm0.097}$ | $0.14_{\pm0.48}$ | $0.14_{\pm0.12}$ | $0.022_{\pm0.28}$ | $0.16_{\pm0.26}$ | $\mathbf{-0.55}_{\pm0.092}$ |
| R&D ratio $r_t^{RD}$ | $0.049_{\pm0.078}$ | $0.028_{\pm0.081}$ | $-0.16_{\pm0.4}$ | $0.076_{\pm0.1}$ | $0.044_{\pm0.23}$ | $-0.26_{\pm0.22}$ | $\mathbf{0.76}_{\pm0.077}$ |
| capital investment ratio $r_t^I$ | $-0.018_{\pm0.1}$ | $-0.094_{\pm0.11}$ | $-0.78_{\pm0.52}$ | $-0.066_{\pm0.13}$ | $-0.021_{\pm0.3}$ | $\mathbf{-0.97}_{\pm0.28}$ | $-0.19_{\pm0.1}$ |

## H.5 ANALYSIS OF LLMS' DECISION-MAKING PATTERNS

Table H.4 reports the regression coefficients for all economic variables in $\mathcal{E}_t^{(f)}$, which are referenced during decision-making. To reduce noise in the regression results, we excluded variables that appeared likely to exhibit multicollinearity. For example, the number of goods sold $n_t$ has a strong

relationship with sales $S_t$ (as dividing by price $p_t$ produces $n_t$), and it also correlates with demand $D_t$. Additionally, to prevent trends from affecting the regression results, we included a trend variable $t$ in the regression, thereby absorbing any trend effects.

In the figure, variables in **bold** indicate those that are both significant and influential. Here, "significant" (i.e., significantly different from zero) means the absolute coefficient value exceeds twice its standard deviation (2-sigma or more), and "influential" means the absolute coefficient value is greater than 0.30.

In contrast to the advanced LLM discussed in Section 5, less advanced LLMs, i.e., GPT-4.1-nano and GPT-3.5-Turbo (Table H.4) scattered their attention across many variables without rational patterns. For instance, they failed to consider demand-supply gaps when setting price change rates, causing "overpricing" (Section H.6). Although Li et al. (2024a) demonstrated that GPT-3.5-Turbo can produce rational patterns for household decisions, our findings indicate that firm-level strategic decision-making remains a significant challenge for these less advanced LLMs.

Table H.5: Monthly average business performance of different LLMs competing as firm agents in the same market. Variables: $S_t$=“sales”, $V_t$=“inventory”, $n_t$=“units sold”, $p_t$=“price”, $w_t$=“hourly wage”, $C_t$=“Personnel cost”.

| | Total Sales | Consumer Goods | | | | Capital Goods | | | | Funding | Personnel Cost | | Capital Investment | Profit |
|---|---|---|---|---|---|---|---|---|---|---|---|---|---|---|
| | $S_t$ | $S_t^C$ | $V_t^C$ | $n_t^C$ | $p_t^C$ | $S_t^K$ | $V_t^K$ | $n_t^K$ | $p_t^K$ | $F_t$ | $w_t$ | $C_t$ | $\overline{M}_t^I$ | $P_t$ |
| GPT-3.5-Turbo | $4,234 | $1,692 | 6806 | 18.1 | $103 | $2,542 | 7904 | 28.4 | $94.0 | $2,684 | $5.71 | $7,471 | $4,218 | − $3,805 |
| GPT-4.1-nano | $6,332 | $769 | 10122 | 8.11 | $46.5 | $5,563 | 4955 | 73.2 | $57.8 | $2,684 | $4.05 | $5,358 | $6,493 | − $1,958 |
| GPT-4.1 | $38,921 | $12,491 | 4288 | 435 | $38.9 | $26,430 | 1343 | 603 | $51.7 | $2,684 | $5.43 | $7,132 | $23,825 | $11,427 |

Table H.6: Monthly average business performance of GPT-4.1 firm agents when referencing economic variables further back in time. The vertical axis represents the *history length* of the referenced economic variables, e.g., “3” indicates that the LLM referred all the variables from $[t-1, t-2, t-3]$. Variables: $S_t$=“sales”, $V_t$=“inventory”, $n_t$=“units sold”, $p_t$=“price”, $w_t$=“hourly wage”, $C_t$=“Personnel cost”.

| | Total Sales | Consumer Goods | | | | Capital Goods | | | | Funding | Personnel Cost | | Capital Investment | Profit |
|---|---|---|---|---|---|---|---|---|---|---|---|---|---|---|
| | $S_t$ | $S_t^C$ | $V_t^C$ | $n_t^C$ | $p_t^C$ | $S_t^K$ | $V_t^K$ | $n_t^K$ | $p_t^K$ | $F_t$ | $w_t$ | $C_t$ | $\overline{M}_t^I$ | $P_t$ |
| 1 | $9,871$\pm\$2.467$ | $5,107$\pm\$2.622$ | 10297$\pm795$ | 71.1$\pm33.6$ | $72.4$\pm\$2.45$ | $4,764$\pm\$1.026$ | 6356$\pm1217$ | 65.2$\pm17.6$ | $72.9$\pm\$1.82$ | − $265$\pm\$797$ | $3.63$\pm\$2.03$ | $4,315$\pm\$2.412$ | $3,719$\pm\$772$ | $1,969$\pm\$1.211$ |
| 2 | $11,560$\pm\$3.337$ | $6,352$\pm\$3.176$ | 11344$\pm1845$ | 91.8$\pm43.7$ | $72.3$\pm\$2.37$ | $5,208$\pm\$798$ | 5207$\pm1593$ | 72.0$\pm11.6$ | $72.7$\pm\$1.95$ | − $265$\pm\$797$ | $3.72$\pm\$2.16$ | $4,429$\pm\$2.569$ | $4,697$\pm\$790$ | $2,237$\pm\$1.132$ |
| 3 | $10,416$\pm\$3.260$ | $5,616$\pm\$3.168$ | 10044$\pm2476$ | 78.4$\pm43.0$ | $72.4$\pm\$2.25$ | $4,800$\pm\$380$ | 6465$\pm1045$ | 64.9$\pm7.86$ | $72.8$\pm\$1.80$ | − $265$\pm\$797$ | $3.57$\pm\$2.10$ | $4,228$\pm\$2.463$ | $4,930$\pm\$679$ | $1,485$\pm\$1.079$ |
| 4 | $10,527$\pm\$2.636$ | $5,856$\pm\$2.429$ | 9726$\pm1318$ | 79.3$\pm34.3$ | $72.5$\pm\$2.12$ | $4,670$\pm\$1.035$ | 6808$\pm1378$ | 62.1$\pm16.7$ | $72.9$\pm\$1.63$ | − $265$\pm\$797$ | $3.66$\pm\$2.15$ | $4,338$\pm\$2.542$ | $5,250$\pm\$1.514$ | $1,193$\pm\$1.225$ |
| 5 | $10,814$\pm\$3.360$ | $6,015$\pm\$2.588$ | 10105$\pm1216$ | 83.8$\pm35.6$ | $72.7$\pm\$1.95$ | $4,799$\pm\$1.691$ | 6414$\pm1191$ | 64.4$\pm26.0$ | $73.2$\pm\$1.47$ | − $265$\pm\$797$ | $3.66$\pm\$2.08$ | $4,338$\pm\$2.447$ | $5,013$\pm\$1.198$ | $1,631$\pm\$2.069$ |
| 7 | $6,687$\pm\$4.003$ | $3,859$\pm\$3.323$ | 10248$\pm1063$ | 51.4$\pm46.6$ | $72.9$\pm\$1.84$ | $2,828$\pm\$1.591$ | 7664$\pm1467$ | 34.5$\pm20.9$ | $73.2$\pm\$1.38$ | − $265$\pm\$797$ | $3.70$\pm\$2.14$ | $4,391$\pm\$2.541$ | $3,178$\pm\$1.421$ | − $712$\pm\$1.628$ |
| 10 | $5,519$\pm\$4.097$ | $3,022$\pm\$2.720$ | 10198$\pm993$ | 39.3$\pm36.9$ | $73.5$\pm\$1.51$ | $2,498$\pm\$1.690$ | 8006$\pm1368$ | 30.8$\pm22.9$ | $73.9$\pm\$1.26$ | − $265$\pm\$797$ | $3.55$\pm\$1.98$ | $4,226$\pm\$2.349$ | $2,780$\pm\$1.825$ | − $1,409$\pm\$1.631$ |

## H.6 CAN FIRM-LLMS ACHIEVE GOOD BUSINESS PERFORMANCE?

To show that LLMs can model the strategic decision-making of real-world firms, we measure the business performance of LLMs as firm agents. We placed multiple LLMs in the same simulation run—forcing them to compete—to determine whether strategic decision-making is unique to advanced models.

Table H.5 shows the business performance of several LLMs: GPT-4.1, a state-of-the-art model; its smaller (inferior) version, GPT-4.1-nano; and an older version, GPT-3.5. GPT-4.1 achieved positive profits ($P_t > 0$), succeeding in business management. In contrast, others operated at a loss ($P_t < 0$).

Profit $P_t$ eq. (17) decomposes as: $P_t = \overline{M}_t - \overline{M}_t^I = S_t + F_t - C_t - \overline{M}_t^I$. Funding $F_t$ and personnel costs $C_t$ were similar among LLMs, and capital investment $\overline{M}_t^I$ just corresponded to sales $S_t$. Therefore, profit differences primarily came from sales $S_t$. Sales consist of consumer and capital goods ($S_t = S_t^C + S_t^K$), and both of the other LLMs lagged behind GPT-4.1's. Alghough inventories ($V_t^C$, $V_t^K$) are sufficient, fewer units ($n_t^C$, $n_t^K$) were sold. Higher prices ($p_t^C$, $p_t^K$) explain the reason: in EconGrowthAgent, only price differentiates goods produced by different firms, so charging more than competitors leads to unsold goods.

Overall, these results suggests that state-of-the-art LLMs (GPT-4.1) can model strategic decision-making in firms, whereas less advanced LLMs cannot.

## H.7 RESULTS WHEN LLMS REFER TO ECONOMIC VARIABLES OVER A LONGER SPAN

Table H.6 presents the business performance of GPT-4.1 agents when referencing economic variables further back in time. Profits $P_t$ remained roughly the same (within a margin of error) for a history length up to five, but decreased when it exceeded that threshold.

We attribute this performance drop to two main reasons:

1. In EconGrowthAgent's market competition, the most critical information comes from the immediate past ($t-1$). Specifically, if a firm sets its price below the other firms' most recent prices, it will likely capture their demand. Hence, a long history is not very helpful for boosting profits.

2. LLMs still struggle to filter out relevant information from large volumes of irrelevant information in a prompt (Razeghi et al., 2022; Shi et al., 2023; Zhang et al., 2024; Mirzadeh et al., 2024; Srivastava et al., 2024; Zhou et al., 2024; Jitsev, 2024; Gulati et al., 2024). A longer history may therefore have confused the LLM, degrading its decision-making.

This second point is particularly significant, suggesting that current LLMs may still struggle to incorporate longer-term economic trends.

