# OpenReview forum: "EconGrowthAgent: Economic Growth Simulation based on LLM Agents and Growth Theory"
_ICLR.cc/2026/Conference — ICLR 2026 Conference Withdrawn Submission_

### Official Review · Reviewer_hmUj · 2025-10-22

**Soundness:** 2
**Presentation:** 2
**Contribution:** 2
**Rating:** 2
**Confidence:** 4

**Summary:**

In this paper, the authors introduce EconGrowthAgent—the first LLM-based ABM for simulating economic growth. It translates macroeconomic growth theory into micro-level interactions among 100 LLM-driven agents over 25 years. The model replicates growth patterns, supports micro-to-macro analysis, and can explore scenarios impossible in reality, such as a civilization-ending asteroid. EconGrowthAgent offers a new computational “laboratory” for economic experimentation and deeper insights into economic dynamics.

**Strengths:**

1. The authors combine LLM-based modeling of economic agents with a novel dynamic framework, providing a detailed micro-level decomposition of economic growth theory in which agents’ decisions and key economic variables interact and evolve over time.
2. Through long-term simulations involving numerous agents, the authors demonstrate that EconGrowthAgent reproduces economic growth and related macroeconomic phenomena, while its micro-level decision-making closely approximates real-world human behavior. Integrated micro-to-macro analyses confirm the model’s validity from both perspectives.
3. EconGrowthAgent enables the exploration of counterfactual scenarios—such as events that are impossible or impractical to test in reality—offering a powerful tool for advancing economic research.

**Weaknesses:**

1. Over-Simplified Simulation Approach: The simulation appears overly simplistic, relying primarily on prompt engineering. This approach may fail to capture the complexity and variability inherent in real-world economic systems. Furthermore, the multi-agent framework is insufficiently described, reducing the transparency and credibility of the simulation design.
2. Lack of Causal and Factor Analysis: The experiments primarily replicate observed patterns of economic growth without investigating the underlying causes or contributing factors. In social simulations, understanding the drivers and mechanisms behind the phenomena is essential, and the absence of such analysis limits the model’s explanatory power.
3. Insufficient Analysis in Counterfactual Scenarios: In Section 6, the paper presents a counterfactual simulation using EconGrowthAgent, but provides minimal analytical discussion. A deeper examination of the results, including interpretation of economic dynamics and implications, would enhance the contribution and rigor of the study.

**Questions:**

See the weaknesses.
More questions:
1. The authors state that “economic growth theory explains growth via two factors.” However, it is not clear whether this theoretical framing remains current and widely accepted in contemporary economic research. The paper would benefit from clarifying the theoretical background and discussing its relevance in relation to more recent developments in growth theory.
2. Related to the above point, similar questions arise in Sections 3, 4, and 5, where micro- and macro-level analyses are presented. The authors could strengthen these sections by citing recent literature to either support their focus on these factors or explain why they remain important and appropriate for the current simulation study. This would enhance the study’s credibility and situate it more firmly within up-to-date scholarly discourse.

---

### Official Review · Reviewer_Goib · 2025-10-30

**Soundness:** 2
**Presentation:** 2
**Contribution:** 1
**Rating:** 2
**Confidence:** 5

**Summary:**

This paper addresses the long-standing challenge of simulating sustained economic growth with agent-based models without hand-crafted decision rules. It decomposes growth theory into a micro-level dynamic system and instantiates households and firms as LLM agents that set labor, consumption, prices, investment, and R&D while interacting through markets. In 25-year simulations with 100 agents, the system reports positive growth driven by capital accumulation and technology, reproduces several qualitative regularities, and runs counterfactual “asteroid shock” experiments.

**Strengths:**

1. The problem formulation is concrete and focused on growth, moving beyond earlier LLM-ABMs that mainly targeted correlations like Phillips/Okun without capital or technology. By tying the micro decisions to Solow/Romer-style factors, the paper offers a clear target for evaluation even if the mapping remains stylized.
2. The micro-to-macro decomposition is explicitly written as state-update equations for production, pricing, investment, and R&D, which makes the simulator easier to reason about than purely prompt-driven worlds. While standard, the explicit separation between consumer and capital goods and between production and R&D labor is helpful for abstractions.
3. The authors attempt to validate firm behavior statistically, regressing LLM decisions on referenced variables and highlighting coefficients consistent with price adjustment to demand–supply gaps and performance-linked wages. This moves beyond screenshots of traces and begins to quantify decision rules.

**Weaknesses:**

1. Novelty is limited, and the methodological delta over existing ABMs and LLM-ABMs is unclear. Core modeling choices—Cobb–Douglas production, capital accumulation, a Romer-style R&D channel, and prompt-mediated pricing/investment—are standard; the paper does not isolate which new component is necessary or sufficient for growth beyond adding firms and capital goods to prior household-only setups. A component-outcome ablation (e.g., remove R&D, fix investment ratios, replace LLM with scripted policies) is needed to establish originality in the mechanism rather than the scope.
2. The empirical evidence for “reproducing growth and key phenomena” is mostly qualitative and statistically thin. Reported GDP gains and factor decompositions lack uncertainty quantification beyond a handful of seeds, and several claims are justified by short visual trends without formal tests, effect sizes, or power analysis across runs and horizons; even the macro comparisons hinge on parameterized personas rather than exogenous shocks with counterfactuals. For ICLR standards, longer horizons, confidence bands, bootstrap across seeds, and sensitivity to functional forms are required.
3. Construct validity is undermined by heavy reliance on chain-of-thought traces as evidence that agents are “human-like.” The paper quotes internal reasoning to justify behaviors (e.g., pricing and R&D choices), but this is not independently verified against human annotations, blinded judges, or external elasticities, and the same LLM family is used as both actor and evaluator in parts of the analysis. Without human-grounded checks or out-of-sample behavioral targets, the interpretation risks circularity.
4. Key economic mechanisms are hard-coded or simplified in ways that can drive the results, weakening the claim of emergence. Capital accumulation mechanically raises output under Cobb–Douglas, R&D augments productivity via a fixed functional form, and firms buy from the lowest-price supplier while wages and prices are revised via direct prompts; the model omits matching frictions, contract structure, inventories with costs, and financial constraints beyond a basic bank. The reported growth could thus be a predictable outcome of the transition path in the specified system rather than an emergent macro regularity of agent behavior.
5. Reproducibility and credibility suffer from dependence on proprietary LLMs, small-N experimental design, and placeholder citations. The main runs use Claude-3.5/GPT-4.1 with approximate costs of about 70 USD per 300-step run and only five seeds; prompts and hyperparameters are only partially specified.

**Questions:**

Please refer to the weakness.

---

### Official Review · Reviewer_CQbH · 2025-11-01

**Soundness:** 3
**Presentation:** 4
**Contribution:** 3
**Rating:** 6
**Confidence:** 4

**Summary:**

This work presents an LLM-based agent-based model that simulates economic growth. In doing so, the authors propose micro-level (company and household) dynamic models where LLM-based agents make economic decisions. In the experiments, the authors demonstrate that the simulated behavior aligns with macroeconomic phenomena and allows micro analysis. A further counterfactual simulation is conducted to verify the model’s capacity for modeling human behavior.

**Strengths:**

Contribution: A reasonable contribution. It extends prior frameworks such as EconAgent https://arxiv.org/abs/2310.10436 by introducing an additional structural layer of entities: firms and households. This represents a step toward a multi-level architecture that is closer to traditional agent-based modeling (ABM). The hierarchical design is an important conceptual advance beyond systems composed only of atomic individuals and opens the door to richer macroeconomic simulations.

Presentation: The paper is well written, clearly organized, and provides sufficient detail.

Experiments: The experiments appear sound. They successfully demonstrate economic growth, and the prompts and dialogue transcripts are coherent. The availability of code is a positive aspect that enhances transparency and reproducibility.

Overall: A clear and well-executed piece of work that advances the integration of LLM agents into economic simulation. It is a reasonable and promising contribution.

**Weaknesses:**

There are several gaps that limit the strength of the paper, which are important for a study focusing on a simulation-based approach to economic behaviors. These issues are best viewed as missing analyses that, if addressed, would substantially strengthen the work.

Framing from an economics perspective: The economic framing remains underspecified. It is unclear why reproducing growth is treated as the main criterion of success, rather than also examining stagnation, decline, or crises, which are equally common in real economies and which a scientific model should also be able to represent. It is uncertain whether the observed growth results from explicit economic mechanisms or from inductive biases in the LLM’s model (LLM can encode social priors, like a tendency to cooperative and rational choices). Clarifying this point and providing a stronger theoretical motivation or economic interpretation of the results would significantly enhance the paper’s significance.

Perspective from complex systems: The analysis would benefit from a more thorough investigation of system sensitivity, which is a core objective of simulation-based research. Specifically:
- Sensitivity analysis: The authors could examine how perturbations affect macro-level outcomes. Relevant perturbations include: (a) parameter sensitivity (as parameters in prompt, like population size,  policy variables), (b) prompt sensitivity (different phrasing or decision heuristics), and (c) initialization or randomness sensitivity (alternative endowment of each agent). Such tests would demonstrate whether the emergent dynamics are robust or fragile.
- Counterfactual and tail analysis: The single example provided ("a giant asteroid approaches Earth") appears arbitrary and insufficient as a stress test. More structured counterfactuals such as policy changes (interest rate), productivity shocks, or out-of-distribution behavioral scenarios would be more informative. Controlled perturbations and combined shocks would clarify when and why the model diverges from a normal trajectory.

Perspective from LLM-based agents:
Because the emergent behavior arises from LLMs, the reliability and consistency of agent responses are critical. Although the paper conducts ten simulation runs per case, it does not evaluate whether the LLM agents behave stably across sufficiently different model backends (only GPT-4.1 and Claude-3.5 Sonnet are used). It also does not provide sufficient evidence regarding the robustness or consistency of the LLM-generated behaviors. Including such analysis would help distinguish between outcomes driven by LLM priors and those that genuinely emerge from economic mechanisms.

**Questions:**

(See unclarities mentioned in Weaknesses)

---

### Official Review · Reviewer_2T2N · 2025-11-02

**Soundness:** 2
**Presentation:** 1
**Contribution:** 1
**Rating:** 2
**Confidence:** 5

**Summary:**

This paper presents EconGrowthAgent, an agent-based model that uses LLMs to simulate economic growth. The authors decompose macroeconomic growth theory into micro-level dynamics where agents make decisions. Through 25-year simulations, the system "reproduces" GDP growth and validates classical economic phenomena including capital accumulation, technological innovation, and the relationship between savings rates and growth. The authors also demonstrate a counterfactual scenario involving an asteroid threat.

**Strengths:**

S1. Economic growth is an important research subject in economics, and using LLM-based ABM is an appropriate modeling approach.

S2. Many settings in the paper are based on existing classical economic theories (Section 2), making the experiments reasonably grounded.

**Weaknesses:**

W1. From an economics perspective, the market mechanisms in this work are extremely simplified. All goods are reduced to two homogeneous categories: consumer goods and capital goods, with purchasing decisions based solely on lowest price. Under this setting, the advantages of LLMs such as domain knowledge and human-like priors, which are commonly leveraged in similar works, are significantly weakened. The difference between using LLMs versus traditional rule-based agents may not be substantial. If such simplifications are necessary, the authors should clearly justify the reasons in the main text, conduct ablation experiments replacing LLM components with rules, and explicitly discuss potential limitations to clarify the scope of their conclusions.

W2. The validation of simulation results in Section 5 lacks rigorous evidence. In line 261, the authors claim to "reproduce" economic growth, which is an overstatement. The two phenomena mentioned in lines 262-263 represent only a small subset of economic growth, and reproducing these patterns does not demonstrate successful reproduction of the phenomenon itself. In Section 5.1, the authors mention an "annual growth rate of about 2.3%" without clarifying what real-world data this corresponds to or why it aligns with reality. Additionally, at line 301, do the authors claim that only kappa and alpha are the core drivers of economic growth? Is this an oversimplification? Section 5.1 seems to compare only with Li et al. (2024a), and I question whether this comparison is reasonable and valid. The authors should justify what is being compared, as Li's work has different core objectives and settings (lacking growth mechanisms) and may not be directly comparable. In the micro-level analysis, judging decision rationality solely through text examples is insufficient. There is no quantitative evidence supporting the validity of the experiments.

W3. From the LLM usage perspective, Table H.4 shows that weaker models perform poorly. Does this indicate the framework is sensitive to model choice? Furthermore, only 100 agents are used, which is hardly an "economy," yet the authors state that cost constraints prevent testing with 300 agents. Does this suggest poor scalability of the framework, making it difficult to simulate microeconomics through bottom-up modeling?

W4. The paper's presentation is unclear, with poor organization of the extensive economic knowledge introduced. Particularly for ICLR readers, it is difficult to distinguish between simplified assumptions and well-established theories strongly aligned with reality. Section 2 introduces existing theories but fails to connect them to how these theories are applied in the model; the theories themselves do not help readers understand the framework's contributions. Section 3 contains too many variables without highlighting which factors are critical. Some formula definitions should be moved to the appendix, focusing instead on explaining the meaning of variables in the model and how variable settings are validated (such as sensitivity analysis). Some text is too small to read (especially Table A.2, requiring 200% zoom). Too much content is relegated to appendices, disrupting the flow of the main text.

W5. This work lacks novel contributions in both economics and AI. Since the rise of LLM-based simulations, there have been many applications of LLMs in economics. This work merely applies similar methods to the phenomenon of economic growth without answering new economic questions or providing insights into how such frameworks can address broader problems. Some core conclusions, such as frugal countries experiencing faster GDP growth, are not novel. The counterfactual experiment is interesting but has limited economic value. Additionally, I find Section 7's discussion of research extensions quite abrupt within the overall paper structure, lacking feasibility analysis and connection to the proposed methods. This should be summarized as future work. Overall, I believe the scope of this work may not be suitable for AI-related conferences; economics or computational economics conferences/journals might be better venues.

**Questions:**

Please see the suggestions above.

---

### Note · Authors · 2025-12-01

**Comment:**

After carefully reflecting on the reviews and the current state of our work, we have realized that the paper requires substantial improvements before it can be ready for publication. We have thoroughly read all of your comments, and we truly appreciate how constructive and insightful they are. We will make sure to use this feedback to guide the next iteration of our work.

Thank you very much for the time and effort you dedicated to reviewing our paper.

**Withdrawal Confirmation:**

I have read and agree with the venue's withdrawal policy on behalf of myself and my co-authors.